# Environment-Aware Dynamic Graph Learning for Out-of-Distribution Generalization

**Haonan Yuan**[1,2], **Qingyun Sun**[1,2],***Xingcheng Fu**[3], **Ziwei Zhang**[4], **Cheng Ji**[1,2],
**Hao Peng**[1], **Jianxin Li**[1,2]

[1]Beijing Advanced Innovation Center for Big Data and Brain Computing, Beihang University
[2]School of Computer Science and Engineering, Beihang University
[3]Key Lab of Education Blockchain and Intelligent Technology, Guangxi Normal University
[4]Department of Computer Science and Technology, Tsinghua University
{yuanhn,sunqy,jicheng,penghao,lijx}@buaa.edu.cn
fuxc@gxnu.edu.cn, zwzhang@tsinghua.edu.cn

## Abstract

Dynamic graph neural networks (DGNNs) are increasingly pervasive in exploiting spatio-temporal patterns on dynamic graphs. However, existing works fail to generalize under distribution shifts, which are common in real-world scenarios. As the generation of dynamic graphs is heavily influenced by latent environments, investigating their impacts on the out-of-distribution (OOD) generalization is critical. However, it remains unexplored with the following two major challenges: **(1)** How to properly model and infer the complex environments on dynamic graphs with distribution shifts? **(2)** How to discover invariant patterns given inferred spatio-temporal environments? To solve these challenges, we propose a novel **E**nvironment-**A**ware dynamic **G**raph **LE**arning (**EAGLE**) framework for OOD generalization by modeling complex coupled environments and exploiting spatio-temporal invariant patterns. Specifically, we first design the environment-aware EA-DGNN to model environments by multi-channel environments disentangling. Then, we propose an environment instantiation mechanism for environment diversification with inferred distributions. Finally, we discriminate spatio-temporal invariant patterns for out-of-distribution prediction by the invariant pattern recognition mechanism and perform fine-grained causal interventions node-wisely with a mixture of instantiated environment samples. Experiments on real-world and synthetic dynamic graph datasets demonstrate the superiority of our method against state-of-the-art baselines under distribution shifts. To the best of our knowledge, we are the first to study OOD generalization on dynamic graphs from the environment learning perspective.

## 1 Introduction

Dynamic graphs are ubiquitous in the real world, like social networks [6, 25], financial transaction networks [61, 93], traffic networks [47, 97, 96], *etc.*, where the graph topology evolves as time passes. Due to their complexity in both spatial and temporal correlation patterns, wide-ranging applications such as relation prediction [39, 68], anomaly detection [10, 69], *etc.* are rather challenging. With excellent expressive power, dynamic graph neural networks (DGNNs) achieve outstanding performance in dynamic graph representation learning by combining the merits of graph neural network (GNN) based models and sequential-based models.

---

*Corresponding author

37th Conference on Neural Information Processing Systems (NeurIPS 2023).

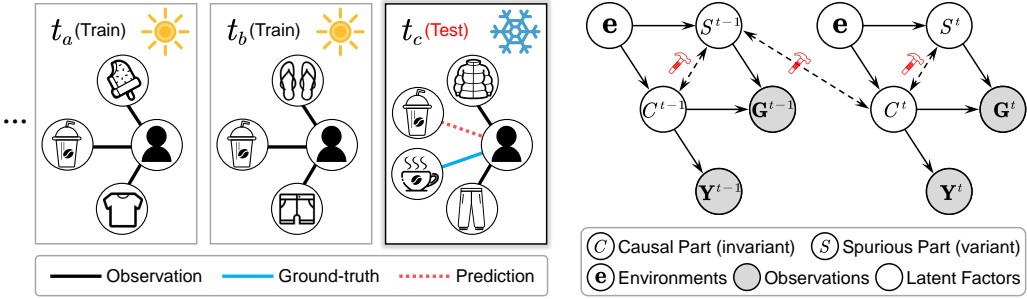

(a) A toy example of distribution shifts on dynamic graphs.  (b) The SCM model on dynamic graphs.

Figure 1: (a) shows the "user-item" interactions with heavy distribution shifts when the underlying environments change (*e.g.*, seasons). A model could mistakenly predict that users would purchase an Iced Americano instead of a Hot Latte if it fails to capture spatio-temporal invariant patterns among spurious correlations. (b) analyzes the underlying cause of distribution shifts on dynamic graphs. It is broadly convinced [64, 67] that the spurious correlations of $C$ (coffee) $\leftrightarrow S$ (cold drink) within and between graph snapshots lead to poor OOD generalization under distribution shifts, which need to be filtered out via carefully investigating the impact of latent **e**. (Detailed discussions in Appendix F.1)

Despite the popularity, most existing works are based on the widely accepted I.I.D. hypothesis, *i.e.*, the training and testing data both follow the independent and identical distribution, which fails to generalize well [77] under the out-of-distribution (OOD) shifts in dynamic scenarios. Figure 1(a) illustrates a toy example demonstrating the OOD shifts on a dynamic graph. The failure lies in: (1) the exploited predictive patterns have changed for in-the-wild extrapolations, causing spurious correlations to be captured [94] and even strengthened with spatio-temporal convolutions. (2) performing causal intervention and inference without considering the impact of environments, introducing external bias of label shifting [91]. These can be explained by SCM model [64, 65, 66] in Figure 1(b).

As the generation of dynamic graphs is influenced by complicated interactions of multiple varying latent environment factors [86, 71, 8, 24, 3, 79, 78, 53], like heterogeneous neighbor relations, multi-community affiliations, *etc.*, investigating the roles environments play in OOD generalization is of paramount importance. To tackle the issues above, we study the problem of generalized representation learning under OOD shifts on dynamic graphs in an environment view by carefully exploiting spatio-temporal invariant patterns. However, this problem is highly challenging in:

- How to understand and model the way latent environments behave on dynamic graphs over time?
- How to infer the distribution of underlying environments with the observed instances?
- How to recognize the spatio-temporal invariant patterns over changing surrounding environments?

To address these challenges, we propose a novel **E**nvironment-**A**ware dynamic **G**raph **LE**arning (**EAGLE**) framework for OOD generalization, which handles OOD generalization problem by exploiting spatio-temporal invariant patterns, thus filtering out spurious correlations via causal interventions over time. Our EAGLE solves the aforementioned challenges and achieves the OOD generalization target as follows. **Firstly**, to shed light on environments, we design the environment-aware EA-DGNN to model environments by multi-channel environments disentangling. It can be considered as learning the disentangled representations under multiple spatio-temporal correlated environments, which makes it possible to infer environment distributions and exploit invariant patterns. **Then**, we propose an environment instantiation mechanism for environment diversification with inferred environment distributions by applying the multi-label variational inference. This endows our EAGLE with higher generalizing power of potential environments. **Lastly**, we propose a novel invariant pattern recognition mechanism for out-of-distribution prediction that satisfies the Invariance Property and Sufficient Condition with theoretical guarantees. We then perform fine-grained causal interventions with a mixture of observed and generated environment instances to minimize the empirical and invariant risks. We optimize the whole framework by training with the adapted min-max strategy, encouraging EAGLE to generalize to diverse environments while learning on the spatio-temporal invariant patterns. Contributions of this paper are summarized as follows:

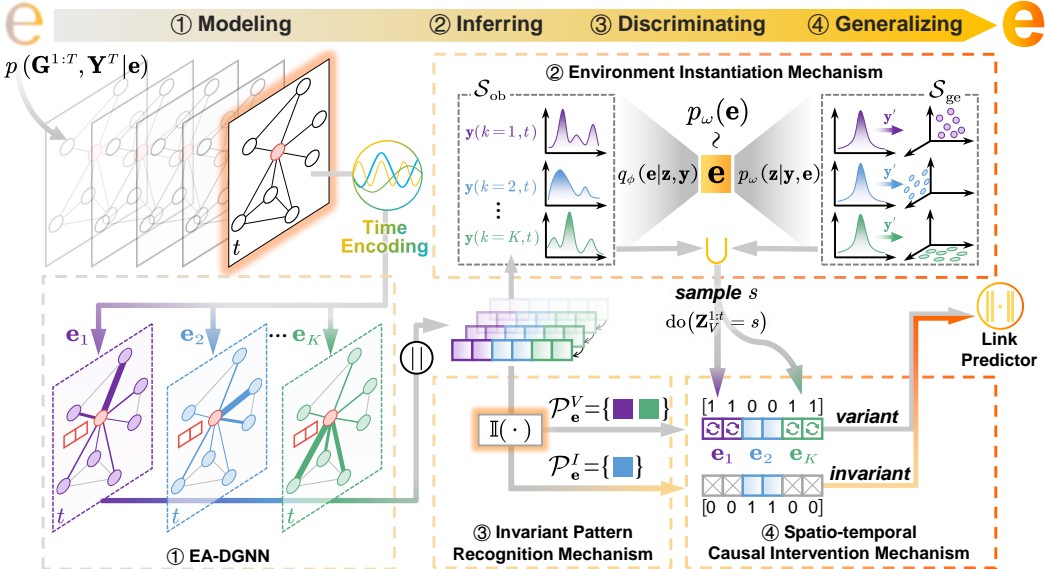

Figure 2: The framework of EAGLE. Our proposed method jointly optimizes the following modules: (1) For a given dynamic graph generated under latent environments, the EA-DGNN first models environment by multi-channel environments disentangling. (2) The environment instantiation mechanism then infers the latent environments, followed by environment diversification with inferred distributions. (3) The invariant pattern recognition mechanism discriminates the spatio-temporal invariant patterns for OOD prediction. (4) Finally, EAGLE generalizes to diverse environments by performing fine-grained causal interventions node-wisely with a mixture of environment instances. The Environment "Modeling-Inferring-Discriminating-Generalizing" paradigm endows EAGLE with higher generalizing power for future potential environments.

- We propose a novel framework named EAGLE for OOD generalization by exploiting spatio-temporal invariant patterns with respect to environments. To the best of our knowledge, this is the first trial to explore the impact of environments on dynamic graphs under OOD shifts.

- We design the Environment "Modeling-Inferring-Discriminating-Generalizing" paradigm to endow EAGLE with higher extrapolation power for future potential environments. EAGLE can carefully model and infer underlying environments, thus recognizing the spatio-temporal invariant patterns and performing fine-grained causal intervention node-wisely with a mixture of observed and generated environment instances, which generalize well to diverse OOD environments.

- Extensive experiments on both real-world and synthetic datasets demonstrate the superiority of our method against state-of-the-art baselines for the future link prediction task.

## 2 Problem Formulation

In this section, we formulate the OOD generalization problem on dynamic graphs. Random variables are denoted as **bold** symbols while their realizations are denoted as *italic* letters. Notation with its description can be found in Appendix A.

**Dynamic Graph Learning.** Denote a dynamic graph as a set of discrete graphs snapshots $\mathbf{DG} = \{\mathcal{G}\}_{t=1}^{T}$, where $T$ is the time length [94, 57], $\mathcal{G}^t = (\mathcal{V}^t, \mathcal{E}^t)$ is the graph snapshot at time $t$, $\mathcal{V}^t$ is the node set and $\mathcal{E}^t$ is the edge set. Let $\mathbf{A}^t \in \{0,1\}^{N \times N}$ be the adjacency matrix and $\mathbf{X}^t \in \mathbb{R}^{N \times d}$ be the matrix of node features, where $N = |\mathcal{V}^t|$ denotes the number of nodes and $d$ denotes the dimensionality. As the most challenging task on dynamic graphs, future link prediction task aims to train a predictor $f_{\boldsymbol{\theta}} : \mathcal{V} \times \mathcal{V} \mapsto \{0,1\}^{N \times N}$ that predicts the existence of edges at time $T+1$ given historical graphs $\mathcal{G}^{1:T}$. Concretely, $f_{\boldsymbol{\theta}} = w \circ g$ is compound of a DGNN $w(\cdot)$ for learning node representations and a link predictor $g(\cdot)$ for predicting links, *i.e.*, $\mathbf{Z}^{1:T} = w(\mathcal{G}^{1:T})$ and $\hat{Y}^T = g(\mathbf{Z}^{1:T})$.

The widely adopted Empirical Risk Minimization (ERM) [82] learns the optimal $f_{\boldsymbol{\theta}}^{\star}$ as follows:

$$\min_{\boldsymbol{\theta}} \mathbb{E}_{(\mathcal{G}^{1:T}, Y^T) \sim p(\mathbf{G}^{1:T}, \mathbf{Y}^T)} \left[ \ell \left( f_{\boldsymbol{\theta}} \left( \mathcal{G}^{1:T} \right), Y^T \right) \right]. \tag{1}$$

**OOD Generalization with Environments.** The predictor $f_{\boldsymbol{\theta}}^{\star}$ learned in Eq. (1) may not generalize well to the testing set when there exist distribution shifts, *i.e.*, $p_{\text{test}}(\mathbf{G}^{1:T}, \mathbf{Y}^T) \neq p_{\text{train}}(\mathbf{G}^{1:T}, \mathbf{Y}^T)$. A broadly accepted assumption is that the distribution shifts on graphs are caused by multiple latent environments $\mathbf{e}$ [86, 71, 8, 24, 3], which influence the generation of graph data, *i.e.*, $p(\mathbf{G}^{1:T}, \mathbf{Y}^T \mid \mathbf{e}) = p(\mathbf{G}^{1:T} \mid \mathbf{e})p(\mathbf{Y}^T \mid \mathbf{G}^{1:T}, \mathbf{e})$. Denoting $\mathbf{E}$ as the support of environments across $T$ time slices, the optimization objective of OOD generalization can be reformulated as:

$$\min_{\boldsymbol{\theta}} \max_{\mathbf{e} \in \mathbf{E}} \mathbb{E}_{(\mathcal{G}^{1:T}, Y^T) \sim p(\mathbf{G}^{1:T}, \mathbf{Y}^T \mid \mathbf{e})} \left[ \ell \left( f_{\boldsymbol{\theta}} \left( \mathcal{G}^{1:T} \right), Y^T \right) \mid \mathbf{e} \right], \tag{2}$$

where the min-max strategy minimizes ERM under the worst possible environment. However, directly optimizing Eq. (2) is infeasible as the environment $\mathbf{e}$ cannot be observed in the data. In addition, the training environments may not cover all the potential environments in practice, *i.e.*, $\mathbf{E}_{\text{train}} \subseteq \mathbf{E}_{\text{test}}$.

# 3 Method

In this section, we introduce **EAGLE**, an **E**nvironment-**A**ware dynamic **G**raph **LE**arning framework, to solve the OOD generalization problem. The overall architecture of EAGLE is shown in Figure 2, following the **Environment "Modeling-Inferring-Discriminating-Generalizing"** paradigm.

## 3.1 Environments Modeling with Environment-aware DGNN

To optimize the objectives of OOD generalization, we first need to infer the environments of dynamic graphs. In practice, the formation of real-world dynamic graphs typically follows a complex process under the impact of underlying environments [56, 51]. For example, the relationships in social networks are usually multi-attribute (*e.g.*, colleagues, classmates, *etc.*) and are changing over time (*e.g.*, past classmates, current colleagues). Consequently, the patterns of message passing and aggregation in both spatial and temporal dimensions should be diversified in different environments, while the existing holistic approaches [94, 52] fail to distinguish them. To model spatio-temporal environments, we propose an **E**nvironment-**A**ware **D**ynamic **G**raph **N**eural **N**etwork (**EA-DGNN**) to exploit environment patterns by multi-channel environments disentangling.

**Environment-aware Convolution Layer (EAConv).** We perform convolutions following the "spatial first, temporal second" paradigm as most DGNNs do [54]. We consider that the occurrence of links is decided by $K$ underlying environments $\mathbf{e}_v = \{\mathbf{e}_k\}_{k=1}^{K}$ of the central node $v$, which can be revealed by the features and structures of local neighbors. To obtain representations under multiple underlying environments, EAConv firstly projects the node features $\mathbf{x}_v^t$ into $K$ subspaces corresponding to different environments. Specifically, the environment-aware representation of node $v$ with respect to the $k$-th environment $\mathbf{e}_k$ at time $t$ is calculated as:

$$\mathbf{z}_{v,k}^t = \sigma \left( \mathbf{W}_k^\top \left( \mathbf{x}_v^t \oplus \text{RTE}(t) \right) + \mathbf{b}_k \right), \tag{3}$$

where $\mathbf{W}_k \in \mathbb{R}^{d \times d'}$ and $\mathbf{b}_k \in \mathbb{R}^{d'}$ are learnable parameters, $\text{RTE}(\cdot)$ is the relative time encoding function, $d'$ is the dimension of $\mathbf{z}_{v,k}^t$, $\sigma(\cdot)$ is the activation function (*e.g.*, the Sigmoid function), and $\oplus$ is the element-wise addition.

To reflect the varying importance of environment patterns under $K$ embedding spaces, we perform multi-channel convolutions with spatio-temporal aggregation reweighting for each $\mathbf{e}_k$. Let $\mathbf{A}_{(u,v),k}^t$ be the weight of edge $(u, v)$ under $\mathbf{e}_k$ at time $t$. The spatial convolutions of one EAConv layer are formulated as (note that we omit the layer superscript for brevity):

$$\hat{\mathbf{z}}_{v,k}^t = \mathbf{z}_{v,k}^t + \sum_{u \in \mathcal{N}^t(v)} \mathbf{A}_{(u,v),k}^t \mathbf{z}_{u,k}^t, \tag{4}$$

$$\mathbf{A}_{(u,v),k}^t = \frac{\exp((\mathbf{z}_{u,k}^t)^\top \mathbf{z}_{v,k}^t)}{\sum_{k'=1}^{K} \exp((\mathbf{z}_{u,k'}^t)^\top \mathbf{z}_{v,k'}^t)}, \tag{5}$$

where $\mathcal{N}^t(v)$ is the neighbors of node $v$ at time $t$ and $\hat{\mathbf{z}}$ denotes the updated node embedding. We then concatenate $K$ embeddings from different environments as the final node representation $\hat{\mathbf{z}}_v^{\mathbf{e},t} = \|_{k=1}^K \hat{\mathbf{z}}_{v,k}^t \in \mathbb{R}^{K \times d'}$, where $\|\cdot$ means concatenation. After that, we conduct temporal convolutions holistically here for graph snapshots at time $t$ and before:

$$\mathbf{z}_v^{\mathbf{e},t} = \frac{1}{t} \sum_{\tau=1}^t \hat{\mathbf{z}}_v^{\mathbf{e},\tau} \in \mathbb{R}^{K \times d'}, \quad \text{where} \quad \hat{\mathbf{z}}_v^{\mathbf{e},\tau} = \left[ \hat{\mathbf{z}}_{v,1}^\tau \| \hat{\mathbf{z}}_{v,2}^\tau \| \cdots \| \hat{\mathbf{z}}_{v,K}^\tau \right]. \tag{6}$$

Note that, the temporal convolutions can be easily extended to other sequential convolution models or compounded with any attention/reweighting mechanisms.

**EA-DGNN Architecture.** The overall architecture of EA-DGNN is similar to general GNNs by stacking $L$ layers of EAConv. In a global view, we exploit the patterns of environments by reweighting the multi-attribute link features with $K$ channels. In the view of an individual node, multiple attributes in the link with neighbors can be disentangled to different attention when carrying out convolutions on both spatial and temporal dimensions. We further model the environment-aware representations by combining node representations at each time snapshot $\mathbf{z}_v^{\mathbf{e}} = \bigcup_{t=1}^T \{\mathbf{z}_v^{\mathbf{e},t}\} \in \mathbb{R}^{T \times (K \times d')}$.

### 3.2 Environments Inferring with Environment Instantiation Mechanism

To infer environments with the environmental patterns, we propose an environment instantiation mechanism with multi-label variational inference, which can generate environment instances given multi-labels of time $t$ and environment index $k$.

**Environment Instantiation Mechanism** is a type of data augmentation to help OOD generalization. We regard the learned node embeddings $\mathbf{z}_{v,k}^t$ as environment samples drawn from the ground-truth distribution of latent environment $\mathbf{e}$. Particularly, we optimize a multi-label **E**nvironment-aware **C**onditional **V**ariational **A**uto**E**ncoder (**ECVAE**) to infer the distribution of $\mathbf{e} \sim q_\phi(\mathbf{e} \mid \mathbf{z}, \mathbf{y})$ across $T$. We have the following proposition. Proof for Proposition 1 can be found in Appendix C.1.

**Proposition 1.** Given observed environment samples from the dynamic graph $\mathcal{G}^{1:T}$ denoted as

$$\mathbf{z} = \bigcup_{v \in \mathcal{V}} \bigcup_{k=1}^K \bigcup_{t=1}^T \{\mathbf{z}_{v,k}^t\} \in \mathbb{R}^{(|\mathcal{V}| \times K \times T) \times d'} \overset{\text{def}}{=\!=\!=} \mathcal{S}_{\text{ob}} \tag{7}$$

with their corresponding one-hot multi-labels $\mathbf{y}$, the environment variable $\mathbf{e}$ is drawn from the prior distribution $p_\omega(\mathbf{e} \mid \mathbf{y})$ across $T$ time slices, and $\mathbf{z}$ is generated from the distribution $p_\omega(\mathbf{z} \mid \mathbf{y}, \mathbf{e})$. Maximizing the conditional log-likelihood $\log p_\omega(\mathbf{z} \mid \mathbf{y})$ leads to an optimal ECVAE by minimizing:

$$\mathcal{L}_{\text{ECVAE}} = \text{KL}[q_\phi(\mathbf{e} \mid \mathbf{z}, \mathbf{y}) \| p_\omega(\mathbf{e} \mid \mathbf{y})] - \frac{1}{|\mathbf{z}|} \sum_{i=1}^{|\mathbf{z}|} \log p_\omega(\mathbf{z} \mid \mathbf{y}, \mathbf{e}^{(i)}), \tag{8}$$

where $\text{KL}[\cdot\|\cdot]$ is the Kullback-Leibler (KL) divergence [37], $|\mathbf{z}|$ is the number of observed environment samples, $\mathbf{e}^{(i)}$ is the $i$-th sampling by the reparameterization trick.

**Sampling and Generating.** Proposition 1 demonstrates the training objectives of the proposed ECVAE. We realize ECVAE using fully connected layers, including an environment recognition network $q_\phi(\mathbf{e} \mid \mathbf{z}, \mathbf{y})$ as the encoder, a conditional prior network $p_\omega(\mathbf{e} \mid \mathbf{y})$ as observed environment instantiations, and an environment sample generation network $p_\omega(\mathbf{z} \mid \mathbf{y}, \mathbf{e})$ as the decoder. By explicitly sampling latent $\mathbf{e}_i$ from $p_\omega(\mathbf{e})$, we can instantiate environments by generating samples $\mathbf{z}_{k_i}^{t_i}$ from the the inferred distribution $p_\omega(\mathbf{z} \mid \mathbf{y}_i, \mathbf{e}_i)$ with any given one-hot multi-label $\mathbf{y}_i$ mixed up with the environment index $k_i$ and time index $t_i$. We denote the generated samples as $\mathcal{S}_{\text{ge}}$, which can greatly expand the diversity of the observed environments $\mathcal{S}_{\text{ob}}$ in Eq. (7).

### 3.3 Environments Discriminating with Invariant Pattern Recognition

Inspired by the Independent Causal Mechanism (ICM) assumption [64, 65, 67], we next propose to learn the spatio-temporal invariant patterns utilizing our inferred and instantiated environments.

**Invariant Pattern Recognition Mechanism.** To encourage the model to rely on the invariant correlation that can generalize under distribution shifts, we propose an invariant pattern recognition mechanism to exploit spatio-temporal invariant patterns. We make the following assumption.

**Assumption 1** (Invariant and Sufficient). Given the dynamic graph $\mathcal{G}^{1:T}$, each node is associated with $K$ surrounding environments. There exist spatio-temporal invariant patterns that can lead to generalized out-of-distribution prediction across all time slices. The function $\mathbb{I}^\star(\cdot)$ can recognize the spatio-temporal invariant patterns $\mathcal{P}_\mathbf{e}^I$ and variant patterns $\mathcal{P}_\mathbf{e}^V$ node-wisely, satisfying:

**(a) Invariance Property**: $\forall \mathbf{e} \in \mathbf{E}$, $\mathbb{I}^\star(\mathbf{Z}^{1:T}) \models \mathcal{P}_\mathbf{e}^I$, *s.t.* $p(\mathbf{Y}^T \mid \mathcal{P}_\mathbf{e}^I, \mathbf{e}) = p(\mathbf{Y}^T \mid \mathcal{P}_\mathbf{e}^I)$;

**(b) Sufficient Condition**: $\mathbf{Y}^T = g(\mathbf{Z}_I^{1:T}) + \epsilon$, *i.e.*, $\mathbf{Y}^T \per\!\!\!\perp \mathcal{P}_\mathbf{e}^V \mid \mathcal{P}_\mathbf{e}^I$, where $\mathbf{Z}_I^{1:T}$ denotes representations under $\mathcal{P}_\mathbf{e}^I$ over time, $g(\cdot)$ is the link predictor, $\epsilon$ is an independent noise.

Assumption 1 indicates that the modeled environment-aware representations of nodes under $\mathcal{P}_\mathbf{e}^I$ contain a portion of spatio-temporal invariant causal features that contribute to generalized OOD prediction over time. Invariance Property guarantees for any environment $\mathbf{e}$, function $\mathbb{I}^\star(\cdot)$ can always recognize the invariant patterns $\mathcal{P}_\mathbf{e}^I$ with the observation of $\mathbf{Z}^{1:T}$, and Sufficient Condition ensures the information contained in $\mathbf{Z}_I^{1:T}$ is adequate for the link predictor to make correct predictions.

Next, we show that $\mathbb{I}^\star(\cdot)$ in Proposition 2 can be instantiated with the following proposition.

**Proposition 2** (A Solution for $\mathbb{I}^\star(\cdot)$). Denote $\mathbf{z}_v^{\mathbf{e}'} = [\mathbf{z}'_{v,1}, \mathbf{z}'_{v,2}, \cdots, \mathbf{z}'_{v,K}]$, where $\mathbf{z}'_{v,k} = \bigcup_{t=1}^T \mathbf{z}_{v,k}^t$. Let $\mathrm{Var}(\mathbf{z}_v^{\mathbf{e}'}) \in \mathbb{R}^K$ represents the variance of $K$ environment-aware representations. The Boolean function $\mathbb{I}(\cdot)$ is a solution for $\mathbb{I}^\star(\cdot)$ with the following state update equation:

$$\mathbb{I}(i,j) = \begin{cases} \mathbb{I}(i-1,j) \vee \mathbb{I}(i-1,j - \mathrm{Var}(\mathbf{z}_v^{\mathbf{e}'})[i-1]), & j \geq \mathrm{Var}(\mathbf{z}_v^{\mathbf{e}'})[i-1] \\ \mathbb{I}(i-1,j), & \text{otherwise} \end{cases}, \quad (9)$$

$\mathbb{I}(i,j)$ indicates whether it is feasible to select from the first $i$ elements in $\mathrm{Var}(\mathbf{z}_v^{\mathbf{e}'})$ so that their sum is $j$. Traversing $j$ in reverse order from $\lfloor \sum \mathrm{Var}(\mathbf{z}_v^{\mathbf{e}'})/2 \rfloor$ until satisfying $\mathbb{I}(K,j)$ is True, we reach:

$$\delta_v = \sum \mathrm{Var}(\mathbf{z}_v^{\mathbf{e}'}) - 2j, \quad (10)$$

$$\mathcal{P}_\mathbf{e}^I(v) = \left\{ \mathbf{e}_k \mid \mathrm{Var}(\mathbf{z}_v^{\mathbf{e}'})[k] \leq \frac{1}{K} \sum \mathrm{Var}(\mathbf{z}_v^{\mathbf{e}'}) - \frac{\delta_v}{2} \right\}, \quad \mathcal{P}_\mathbf{e}^V(v) = \overline{\mathcal{P}_\mathbf{e}^I(v)}, \quad (11)$$

where $\delta_v$ is the optimal spatio-temporal invariance threshold of node $v$.

Proposition 2 solves a dynamic programming problem to obtain the optimal $\mathbb{I}^\star(\cdot)$, whose target is to find a partition dividing all patterns into variant/invariant set with the maximum difference between the variance means that reveals the current invariance status, providing an ideal optimizing start point. The proof can be found in Appendix C.2.

With the theoretical support of Assumption 1 and Proposition 2, we can recognize the spatio-temporal invariant/variant patterns $\mathcal{P}_\mathbf{e}^I = \bigcup_{v \in \mathcal{V}} \mathcal{P}_\mathbf{e}^I(v)$ and $\mathcal{P}_\mathbf{e}^V = \bigcup_{v \in \mathcal{V}} \mathcal{P}_\mathbf{e}^V(v)$ with respect to the underlying environments for each node over a period of time by applying $\mathbb{I}(\cdot)$.

### 3.4 Environments Generalizing with Causal Intervention

**Generalization Optimization Objective.** Eq. (2) clarifies the training objective of OOD generalization. However, directly optimizing Eq. (2) is not impracticable. Based on the inferred and generated environments in Section 3.2 and spatio-temporal invariant patterns learned in Section 3.3, we further modify Eq. (2) to improve the model's OOD generalization ability by applying the intervention mechanism node-wisely with causal inference. Specifically, we have the following objectives:

$$\min_{\boldsymbol{\theta}} \mathcal{L}_{\text{task}} + \alpha \mathcal{L}_{\text{risk}}, \quad (12)$$

$$\mathcal{L}_{\text{task}} = \mathbb{E}_{\mathbf{e} \sim q_\phi(\mathbf{e}), (\mathcal{G}^{1:T}, Y^T) \sim p(\mathbf{G}^{1:T}, \mathbf{Y}^T | \mathbf{e})} \left[ \ell \left( g(\mathbf{Z}_I^{1:T}), Y^T \right) \right], \quad (13)$$

$$\mathcal{L}_{\text{risk}} = \mathrm{Var}_{s \in \mathcal{S}} \left\{ \mathbb{E}_{\mathbf{e} \sim q_\phi(\mathbf{e}), (\mathcal{G}^{1:T}, Y^T) \sim p(\mathbf{G}^{1:T}, \mathbf{Y}^T | \mathbf{e})} \left[ \ell \left( f_{\boldsymbol{\theta}} \left( \mathcal{G}^{1:T} \right), Y^T \mid \mathrm{do}(\mathbf{Z}_V^{1:T} = s) \right) \right] \right\}, \quad (14)$$

where $q_\phi(\mathbf{e})$ is the environment distribution in Section 3.2, $\mathrm{do}(\cdot)$ is the *do*-calculus that intervenes the variant patterns, $\mathcal{S} = \mathcal{S}_{\text{ob}} \cup \mathcal{S}_{\text{ge}}$ is the observed and generated environment instances for interventions, and $\alpha$ is a hyperparameter. We show the optimality of Eq. (12) with the following propositions:

**Proposition 3** (Achievable Assumption). Minimizing Eq. (12) can encourage the model to satisfy the Invariance Property and Sufficient Condition in Assumption 1.

**Proposition 4** (Equivalent Optimization). Optimizing Eq. (12) is equivalent to minimizing the upper bound of the OOD generalization error in Eq. (2).

Proposition 3 avoids over-strong assumptions of the Sufficient Condition and Invariance Property. Proposition 4 guarantees the OOD generalization error bound of the learned model is within our expectation, which can also be explained in the perspective of the SCM model on dynamic graphs (Figure 1(b)) that, optimizing Eq. (12) eliminates the negative impacts brought by spatio-temporal spurious correlations between labels and variant patterns, while strengthening the invariant causal correlations under diverse latent environments. Proofs for Proposition 3 and Proposition 4 can be found in Appendix C.3 and C.4, respectively.

**Spatio-temporal Causal Intervention Mechanism.** As directly intervening the variant patterns node-wisely is time-consuming and intractable, we propose to approximately perform fine-grained spatio-temporal interventions with $\mathcal{S}_{\mathrm{ob}} \cup \mathcal{S}_{\mathrm{ge}}$ by sampling and replacing variant patterns. For environment-aware representations $\mathbf{z}_v^{\mathbf{e}t} = [\mathbf{z}_{v,1}^t, \mathbf{z}_{v,2}^t, \cdots, \mathbf{z}_{v,K}^t]$ of node $v$ at time $t$, we can obtain $\mathcal{P}_{\mathbf{e}}^I(v)$ and $\mathcal{P}_{\mathbf{e}}^V(v)$ by applying $\mathbb{I}(\cdot)$. As $\mathcal{P}_{\mathbf{e}}^V(v)$ represents environment indices related to variant patterns, we randomly sample instances as the intervention set $\mathbf{s}_v$ and perform node-wise replacing:

$$\mathbf{z}_v^{\mathbf{e},t}\left[\mathcal{P}_{\mathbf{e}}^V(v)\right] := \mathbf{s}_v, \quad \mathbf{s}_v = \{s \mid s \in \mathcal{S}_{\mathrm{ob}} \cup \mathcal{S}_{\mathrm{ge}}\}. \tag{15}$$

Note that, for each intervention operation, we randomly perform each replacement with a different $s$. Further considering ECVAE loss in Eq. (8), the overall loss of EAGLE is defined as:

$$\mathcal{L} = \underbrace{\mathcal{L}_{\mathrm{task}}}_{\mathrm{Eq.\ (13)}} + \alpha \underbrace{\mathcal{L}_{\mathrm{risk}}}_{\mathrm{Eq.\ (14)}} + \beta \underbrace{\mathcal{L}_{\mathrm{ECVAE}}}_{\mathrm{Eq.\ (8)}}, \tag{16}$$

where $\beta$ is another hyperparameter. We conduct hyperparameter sensitivity analysis in Appendix D.4. The training process of EAGLE is summarized in Appendix B.

## 4 Experiments

**Datasets.** We use three real-world datasets to evaluate EAGLE[1] on the challenging link prediction task. **COLLAB** [81] is an academic collaboration dataset with papers published in 16 years; **Yelp** [75] contains customer reviews on business for 24 months; **ACT** [45] describes students actions on a MOOC platform within 30 days. Statistics of the three datasets are concluded in Table D.1.

**Baselines.** We compare EAGLE with representative GNNs and OOD generalization methods.

- **Static GNNs**: GAE [42] and VGAE [42] are GCN [41] based autoencoders on static graphs.
- **Dynamic GNNs**: GCRN [76] adopts GCNs to obtain node embeddings, followed by a GRU [13] to capture temporal relations; EvolveGCN [62] applies an LSTM [31] or GRU to evolve the parameters of GCNs; DySAT [75] models self-attentions in both structural and temporal domains.
- **OOD Generalization Methods**: IRM [3] learns an invariant predictor to minimize invariant risk; V-REx [44] extends IRM by reweighting the risk; GroupDRO [74] reduces the risk gap across training distributions; DIDA [94] exploits invariant patterns on dynamic graphs.

### 4.1 Evaluation of Performance on Future Link Prediction Task

#### 4.1.1 Distribution Shifts on Link Attributes

**Settings.** Each dataset owns multi-attribute relations. We filter out one certain attribute links as the shifted variables under the future OOD environments in validation and training sets chronologically. This is more practical and challenging in real-world scenarios as the model cannot get access to any information about the filtered links until testing stages. Note that, all attribute features have been removed after the above operations before feeding to EAGLE. Detailed settings are in Appendix D.3.

**Results.** Results are shown in Table 1. *w/o OOD* and *w/ OOD* denote testing without and with distribution shifts. The performance of all baselines drops dramatically on each dataset when there exist distribution shifts. Static GNN baselines fail in both settings as they cannot model dynamics.

---

[1] Code is available at `https://github.com/RingBDStack/EAGLE` on PyTorch [63] and MindSpore [33].

Though dynamic GNNs are designed to capture spatio-temporal features, they even underperform static GNNs in several settings, mainly because the predictive patterns they exploited are variant with spurious correlations. Conventional OOD generalization baselines have limited improvements as they rely on the environment labels to generalize, which are inaccessible on dynamic graphs. As the most related work, DIDA [94] achieves further progress by learning invariant patterns. However, it is limited by the lack of in-depth analysis into the environments, causing the label shift phenomenon [91] that damages its generalization ability. With the critical investigation of latent environments and theoretical guarantees, our EAGLE consistently outperforms the baselines and achieves the best performance in all *w/ OOD* settings and two *w/o OOD* settings. Especially even on the most challenging COLLAB, where the time span is extremely long (1990 to 2006) and its link attributes difference is huge.

Table 1: AUC score (% ± standard deviation) of future link prediction task on real-world datasets with OOD shifts of link attributes. The best results are shown in **bold** and the runner-ups are underlined.

| Dataset | COLLAB | | Yelp | | ACT | |
|---|---|---|---|---|---|---|
| Model | *w/o OOD* | *w/ OOD* | *w/o OOD* | *w/ OOD* | *w/o OOD* | *w/ OOD* |
| GAE [42] | 77.15±0.50 | 74.04±0.75 | 70.67±1.11 | 64.45±5.02 | 72.31±0.53 | 60.27±0.41 |
| VGAE [42] | 86.47±0.04 | 74.95±1.25 | 76.54±0.50 | 65.33±1.43 | 79.18±0.47 | 66.29±1.33 |
| GCRN [76] | 82.78±0.54 | 69.72±0.45 | 68.59±1.05 | 54.68±7.59 | 76.28±0.51 | 64.35±1.24 |
| EvolveGCN [62] | 86.62±0.95 | 76.15±0.91 | 78.21±0.03 | 53.82±2.06 | 74.55±0.33 | 63.17±1.05 |
| DySAT [75] | 88.77±0.23 | 76.59±0.20 | 78.87±0.57 | 66.09±1.42 | 78.52±0.40 | 66.55±1.21 |
| IRM [3] | 87.96±0.90 | 75.42±0.87 | 66.49±10.78 | 56.02±16.08 | 80.02±0.57 | 69.19±1.35 |
| V-REx [44] | 88.31±0.32 | 76.24±0.77 | 79.04±0.16 | 66.41±1.87 | 83.11±0.29 | 70.15±1.09 |
| GroupDRO [74] | 88.76±0.12 | 76.33±0.29 | 79.38±0.42 | 66.97±0.61 | 85.19±0.53 | 74.35±1.62 |
| DIDA [94] | 91.97±0.05 | 81.87±0.40 | 78.22±0.40 | 75.92±0.90 | 89.84±0.82 | 78.64±0.97 |
| EAGLE | **92.45±0.21** | **84.41±0.87** | 78.97±0.31 | **77.26±0.74** | **92.37±0.53** | **82.70±0.72** |

### 4.1.2 Distribution Shifts of Node Features

**Setting.** We further introduce settings of node feature shifts on COLLAB. We respectively sample $p(t)|\mathcal{E}^{t+1}|$ positive links and $(1 - p(t))|\mathcal{E}^{t+1}|$ negative links, which are then factorized into shifted features $\mathbf{X}^{t\prime} \in \mathbb{R}^{|\mathcal{V}| \times d}$ while preserving structural property. The sampling probability $p(t) = \bar{p} + \sigma \cos(t)$, where $\mathbf{X}^{t\prime}$ with higher $p(t)$ will have stronger spurious correlations with future underlying environments. We set $\bar{p}$ to be 0.4, 0.6 and 0.8 for training and 0.1 for testing. In this way, training data are relatively higher spuriously correlated than the testing data. We omit results on static GNNs as they cannot support dynamic node features. Detailed settings are in Appendix D.3.

**Results.** Results are reported in Table 2. We can observe a similar trend as Table 1, *i.e.*, our method better handles distribution shifts of node features on dynamic graphs than the baselines. Though some baselines can report reasonably high performance on the training set, their performance drops drastically in the testing stage. In comparison, our method reports considerably smaller gaps. In particular, our EAGLE surpasses the best-performed baseline by approximately 3%/5%/10% of AUC in the testing set under different shifting levels. Interestingly, as severer distribution shifts will lead to more significant performance degradation where the spurious correlations are strengthened, our method gains better task performance with a higher shifting degree. This demonstrates EAGLE is more capable of eliminating spatio-temporal spurious correlations in severer OOD environments.

### 4.2 Investigation on Invariant Pattern Recognition Mechanism

**Settings.** We generate synthetic datasets by manipulating environments to investigate the effect of the invariant pattern recognition mechanism. We set $K = 5$ and let $\sigma_{\mathbf{e}}$ represent the proportion of the environments in which the invariant patterns are learned, where higher $\sigma_{\mathbf{e}}$ means more reliable invariant patterns. Node features are drawn from multivariate normal distributions, and features under the invariant patterns related $\mathbf{e}_k$ will be perturbed slightly and vice versa. We construct graph structures and filter out links built under a certain $\mathbf{e}_k$ as the same in Section 4.1.1. We compare EAGLE with the most related and strongest baseline DIDA [94]. Detailed settings are in Appendix D.3.

**Results.** Results are shown in Figure 3. $\mathbb{I}_{\text{ACC}}$ denotes the prediction accuracy of the invariant patterns by $\mathbb{I}(\cdot)$. We observe that, as $\sigma_{\mathbf{e}}$ increases, the performance of EAGLE shows a significant increase from 54.26% to 65.09% while narrowing the gap between *w/o OOD* and *w/ OOD* scenarios.

Table 2: AUC score (% ± standard deviation) of future link prediction task on real-world datasets with OOD shifts of node features. The best results are shown in **bold** and the runner-ups are underlined.

| Dataset | COLLAB ($\bar{p} = 0.4$) | | COLLAB ($\bar{p} = 0.6$) | | COLLAB ($\bar{p} = 0.8$) | |
| --- | --- | --- | --- | --- | --- | --- |
| **Model** | Train | Test | Train | Test | Train | Test |
| GCRN [76] | 69.60±1.14 | 72.57±0.72 | 74.71±0.17 | 72.29±0.47 | 75.69±0.07 | 67.26±0.22 |
| EvolveGCN [62] | 78.82±1.40 | 69.00±0.53 | 79.47±1.68 | 62.70±1.14 | 81.07±4.10 | 60.13±0.89 |
| DySAT [75] | 84.71±0.80 | 70.24±1.26 | 89.77±0.32 | 64.01±0.19 | 94.02±1.29 | 62.19±0.39 |
| IRM [3] | 85.20±0.07 | 69.40±0.09 | 89.48±0.22 | 63.97±0.37 | **95.02±0.09** | 62.66±0.33 |
| V-REx [44] | 84.77±0.84 | 70.44±1.08 | 89.81±0.21 | 63.99±0.21 | 94.06±1.30 | 62.21±0.40 |
| GroupDRO [74] | 84.78±0.85 | 70.30±1.23 | 89.90±0.11 | 64.05±0.21 | 94.08±1.33 | 62.13±0.35 |
| DIDA [94] | 87.92±0.92 | 85.20±0.84 | 91.22±0.59 | 82.89±0.23 | 92.72±2.16 | 72.59±3.31 |
| **EAGLE** | **92.97±0.88** | **88.32±0.61** | **94.52±0.42** | **87.29±0.71** | 94.11±1.03 | **82.30±0.75** |

Though DIDA [94] also shows an upward trend, its growth rate is more gradual. This indicates that, as DIDA [94] is incapable of modeling the environments, it is difficult to perceive changes in the underlying environments caused by different $\sigma_\mathbf{e}$, thus cannot achieve satisfying generalization performance. In comparison, our EAGLE can exploit more reliable invariant patterns, thus performing high-quality invariant learning and efficient causal interventions, and achieving better generalization ability. In addition, we also observe a positive correlation between $\mathbb{I}_{\text{ACC}}$ and the AUC, indicating the improvements is attributed to the accurate recognition of the invariant patterns by $\mathbb{I}(\cdot)$. Additional results are in Appendix D.7.

### 4.3 Ablation Study

In this section, we conduct ablation studies to analyze the effectiveness of three main mechanisms:

- **EAGLE (*w/o EI*).** We remove the **E**nvironment **I**nstantiation mechanism in Section 3.2, and carrying out causal interventions in Eq. (15) with only observed environment samples.
- **EAGLE (*w/o IPR*).** We remove the **I**nvariant **P**attern **R**ecognition mechanism in Section 3.3, and determining the spatio-temporal invariant patterns $\mathcal{P}_\mathbf{e}^I$ by randomly generating $\delta_v$ for each node.
- **EAGLE (*w/o Interv*).** We remove the spatio-temporal causal **Interv**ention mechanism in Section 3.4 and directly optimize the model by Eq. (16) without the second $\mathcal{L}_{\text{risk}}$ term.

**Results.** Results are demonstrated in Figure 4. Overall, EAGLE consistently outperforms the other three variants on all datasets. The ablation studies provide insights into the effectiveness of the proposed mechanisms and demonstrate their importance in achieving better performance for OOD generalization on dynamic graphs.

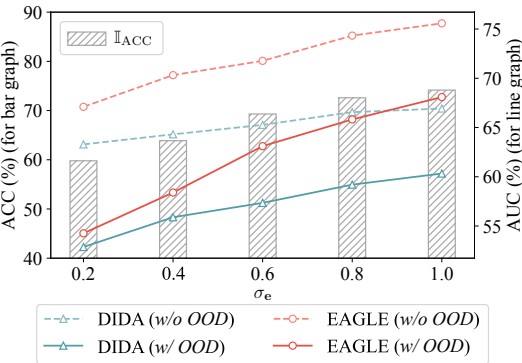

Figure 3: Effects of invariant pattern recognition.

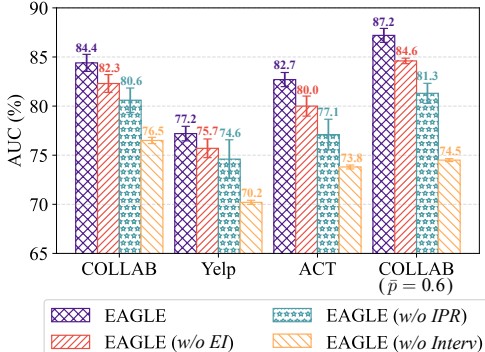

Figure 4: Results of ablation study.

### 4.4 Visualization

We visualize snapshots in COLLAB using NetworkX [27] as shown in Figure 5, where colors reflect latent environments, numbers denote edge weights, solid lines implicate spatio-temporal invariant patterns dependencies, and dashed lines implicate variant patterns dependencies. EAGLE can gradually exploit the optimal invariant patterns and strengthen reliance, making generalized predictions.

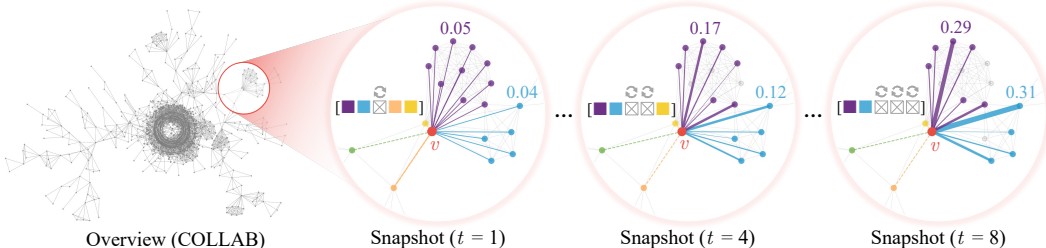

Figure 5: Visualization of snapshots in COLLAB.

## 5 Related Work

**Dynamic Graph Learning.** Extensive researches [70, 4, 38, 90, 85, 35] address the challenges of dynamic graph learning. Dynamic graph neural networks (DGNNs) are intrinsically utilized to model both spatial and temporal patterns [88, 76, 84, 73, 62, 75, 80, 21] by combining vanilla GNNs and sequential-based models [58, 31, 28]. However, most existing works fail to generalize under distribution shifts. DIDA [94] is the sole prior work that tackles distribution shifts on dynamic graphs, but it neglects to model the complex environments, which is crucial in identifying invariant patterns.

**Out-of-Distribution Generalization.** Most machine learning methods are built on the I.I.D. hypothesis, which can hardly be satisfied in real-world scenarios [77, 71, 3]. Out-of-distribution (OOD) generalization has been extensively studied in both academia and industry areas [77, 92, 30] and we mainly focus on graphs. Most works concentrate on static graphs for node-level or graph-level tasks [98, 18, 49, 86, 52, 12, 87], supporting by invariant learning method [14, 46, 95] with disentangled learning [5, 55] and causal inference theories [65, 66, 64]. However, there lack of further research on dynamic graphs with more complicated shift patterns caused by spatio-temporal varying latent environments, which is our main concern.

**Invariant Learning.** Invariant learning aims to exploit the fewer variant patterns that lead to informative and discriminative representations for stable prediction [14, 46, 95]. Supporting by disentangled learning theories and causal learning theories, invariant learning tackles the OOD generalization problem from a more theoretical perspective, revealing a promising power. Disentangle-based methods [5, 55] learn representations by separating semantic factors of variations in data, making it easier to distinguish invariant factors and establish reliable correlations. Causal-based methods [22, 3, 72, 87, 11, 2, 60] utilize Structural Causal Model (SCM) [64] to filter out spurious correlations by intervention or counterfactual with $do$-calculus [65, 66] and strengthen the invariant causal patterns. However, the invariant learning method of node-level tasks on dynamic graphs is underexplored, mainly due to its complexity in analyzing both spatial and temporal invariant patterns.

## 6 Conclusion

In this paper, we propose a novel framework **EAGLE** for out-of-distribution (OOD) generalization on dynamic graphs by modeling complex dynamic environments for the first time and further exploiting spatio-temporal invariant patterns. EAGLE first models environment by an environment-aware DGNN, and then diversifies the observed environment samples with the environment instantiation mechanism, and finally learns OOD generalized spatio-temporal invariant patterns by the invariant pattern recognition mechanism and performs fine-grained causal interventions node-wisely. Experiment results on both real-world and synthetic datasets demonstrate that EAGLE greatly outperforms existing methods under distribution shifts. One limitation is that we mainly consider the node-level tasks, and leave extending our method to the graph-level OOD generalization for future explorations.

## Acknowledgements

The corresponding author is Qingyun Sun. The authors of this paper are supported by the National Natural Science Foundation of China through grants No.62225202, No.62302023, and No.62206149. We owe sincere thanks to all authors for their valuable efforts and contributions. We also acknowledge the support of MindSpore, CANN (Compute Architecture for Neural Networks) and Ascend AI Processor used for this research.

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

## A  Notations

| Notations | Descriptions |
|---|---|
| $\mathbf{DG} = \{\mathcal{G}\}_{t=1}^{T}$ | Dynamic graph (a set of $T$ discrete graph snapshots) |
| $\mathcal{G} = (\mathcal{V}, \mathcal{E})$ | Graph with the node set $\mathcal{V}$ and edge set $\mathcal{E}$ |
| $\mathcal{G}^{t} = (\mathcal{V}^{t}, \mathcal{E}^{t})$ | Graph snapshot at time $t$ |
| $\mathbf{X}^{t}, \mathbf{A}^{t}$ | Node features matrix and adjacency matrix of a graph at time $t$ |
| $\mathbf{x}_{v}^{t}, \mathbf{A}_{(u,v),k}^{t}$ | Node features and edge weights of $(u, v)$ under $\mathbf{e}_{k}$ |
| $\mathcal{G}^{1:t}, Y^{t}, \mathbf{G}^{1:t}, \mathbf{Y}^{t}$ | Graph trajectory, labels and their corresponding random variables |
| $\mathbf{e}, \mathbf{e}_{i}, \mathbf{E}$ | Latent environments and their support |
| $\mathbf{z}_{v,k}^{t}, \mathbf{z}_{v}^{\mathbf{e},t}, \mathbf{z}_{v}^{\mathbf{e}}$ | Node representations under $\mathbf{e}_{k}$, at time $t$ and at overall time slices |
| $\mathbf{z}, \mathbf{y}$ | Observed environment sample with its label |
| $d, d'$ | Dimension for $\mathbf{x}_{v}^{t}$ and $\mathbf{z}_{v,k}^{t}$, respectively |
| $K$ | Number of underlying environments (number of convolution channel) |
| $f(\cdot), w(\cdot), g(\cdot)$ | Model, encoder, and the link predictor |
| $\ell(\cdot)$ | The loss function |
| $q_{\phi}, p_{\omega}$ | The prior distribution and variational distribution of environments |
| $\mathbb{I}^{\star}(\cdot), \mathbb{I}(\cdot)$ | Invariant pattern recognition function and its implementation |
| $\mathcal{P}_{\mathbf{e}}^{I}, \mathcal{P}_{\mathbf{e}}^{V}, \mathcal{P}_{\mathbf{e}}^{I}(v), \mathcal{P}_{\mathbf{e}}^{V}(v)$ | Summary of spatio-temporal invariant/variant patterns for each node |
| $\mathbf{Z}^{1:t}, \mathbf{Z}_{I}^{1:t}, \mathbf{Z}_{V}^{1:t}$ | Summary of node representations, and their variants under $\mathcal{P}_{\mathbf{e}}^{I}$ and $\mathcal{P}_{\mathbf{e}}^{V}$ |
| $\mathcal{S}_{\mathrm{ob}}, \mathcal{S}_{\mathrm{ge}}$ | Observed and generated environments sample libraries |
| $s, \mathbf{s}_{v}$ | Intervention samples from $\mathcal{S}_{\mathrm{ob}} \cup \mathcal{S}_{\mathrm{ge}}$ and their summary for node $v$ |
| $\mathrm{do}(\cdot)$ | *do*-caculus for causal interventions |
| $\mathcal{L}_{\mathrm{task}}, \mathcal{L}_{\mathrm{risk}}, \mathcal{L}_{\mathrm{ECVAE}}$ | Task loss, the invariance loss and the ECVAE loss |
| $\alpha, \beta$ | Hyperparameters for loss trade-off |

## B  Algorithm and Complexity Analysis

---

**Algorithm 1:** Overall training process of EAGLE.

---

**Input:** Dynamic graph $\mathbf{DG} = (\{\mathcal{G}\}_{t=1}^{T})$ with labels $\mathbf{Y}^{1:T}$ of link occurrence; Number of training epochs $E$; Number of intervention times $S$; Hyperparameters $\alpha$ and $\beta$.
**Output:** Optimized model $f_{\boldsymbol{\theta}}^{\star}$; Predicted label $Y^{T}$ of link occurrence at time $T + 1$.

1 Initialize parameters randomly;
2 **for** $i = 1, 2, \cdots, E$ **do**

    // Environments Modeling and Inferring
3     Obtain representations for each node at each time with the support of $\mathbf{e}$, as $\mathbf{z}_{v}^{\mathbf{e},t} \leftarrow$ Eq. (6);
4     Establish the observed environment samples library $\mathcal{S}_{\mathrm{ob}} \leftarrow$ Eq. (7);
5     Infer the distribution $p_{\omega}(\mathbf{e})$ with $\mathcal{L}_{\mathrm{ECVAE}} \leftarrow$ Eq. (8) and generate samples library $\mathcal{S}_{\mathrm{ge}}$;

    // Environments Extrapolating
6     Learn the invariance threshold $\delta_{v} \leftarrow$ Eq. (10) by function $\mathbb{I}(\cdot) \leftarrow$ Eq. (9);
7     Recognize the invariant/variant patterns for each node, as $\mathcal{P}_{\mathbf{e}}^{I}(v), \mathcal{P}_{\mathbf{e}}^{V}(v) \leftarrow$ Eq. (11);
8     Calculate task loss depending on the invariant patterns, as $\mathcal{L}_{\mathrm{task}} \leftarrow$ Eq. (13);

    // Environments Generalizing
9     **for** $j = 1, 2, \cdots, S$ **do**
10         Sample items from $\mathcal{S}_{\mathrm{ob}} \cup \mathcal{S}_{\mathrm{ge}}$ and perform intervention for each node, as Eq. (15);
11         Calculate intervention loss, as $\mathcal{L}_{\mathrm{risk}} \leftarrow$ Eq. (14);
12     **end**

    // Optimize
13     Calculate the overall loss, as $\mathcal{L} \leftarrow$ Eq. (16);
14     Update model parameters by minimizing $\mathcal{L}$.
15 **end**

---

The overall training process of our EAGLE is shown in Algorithm 1.

**Computational Complexity Analysis.** We analyze the computational complexity of each part in EAGLE as follows. Denote $|\mathcal{V}|$ and $|\mathcal{E}|$ as the total number of nodes and edges in each graph snapshot.

In Section 3.1, operations of the EAConv layer in EA-DGNN can be parallelized across all nodes, which is highly efficient. Thus, the computation complexity of EA-DGNN is:

$$\mathcal{O}\left(|\mathcal{E}|\sum_{l=0}^{L}d^{(l)} + \mathcal{V}\left(\sum_{l=1}^{L}d^{(l-1)}d^{(l)} + (d^{(L)})^2\right)\right), \tag{B.1}$$

where $d^{(l)}$ denotes the dimension of the $l$-th layer. As $L$ is a small number, and $d^{(l)}$ is a constant, the Eq. (B.1) can be rewritten as $\mathcal{O}(|\mathcal{E}|d + |\mathcal{V}|d^2)$, where $d$ is the universal notation of all $d^{(l)}$.

In Section 3.2, the computation complexity of the ECVAE is a compound of the encoder and decoder, with the same computation complexity as $\mathcal{O}(|\mathbf{z}|L'd)$, where $|\mathbf{z}|$ is the number of the observed environment samples, $L'$ is the number of layers in the encoder and decoder. Also, as $L'$ is a small number, we omit it for brevity. Thus, the computation complexity of ECVAE is $\mathcal{O}(|\mathbf{z}|d)$.

In Section 3.3, we recognize the invariant/variant patterns for all nodes by the function $\mathbb{I}(\cdot)$ in parallel, with the computation complexity $\mathcal{O}(K\log|\mathcal{V}|)$.

In Section 3.4, we perform sampling and replacing as an implementation of causal interventions. Denote $|\mathcal{E}|_p$ as the number of edges to predict and $|\mathcal{S}|$ as the size of the intervention set, which is usually set as a small constant. The spatio-temporal causal intervention mechanism owns a computation complexity compounding of sampling and replacing as $\mathcal{O}(|\mathcal{S}|d) + \mathcal{O}(|\mathcal{E}_p||\mathcal{S}|d)$ in training, and no extra computation complexity in the inference stage.

Therefore, the overall computation complexity of EAGLE is:
$$\mathcal{O}(|\mathcal{E}|d + |\mathcal{V}|d^2) + \mathcal{O}(|\mathbf{z}|d) + \mathcal{O}(K\log|\mathcal{V}|) + \mathcal{O}(|\mathcal{S}|d) + \mathcal{O}(|\mathcal{E}_p||\mathcal{S}|d). \tag{B.2}$$

In summary, EAGLE has a linear computation complexity with respect to the number of nodes and edges, which is on par with DIDA [94] and other existing dynamic GNNs. We believe that the computational complexity bottleneck of EAGLE lies in the spatio-temporal causal intervention mechanism. We further analyze the intervention efficiency in Appendix D.5.

**Space Complexity Analysis.** We analyze the space complexity of each part in EAGLE as follows. Denote $|\mathcal{V}|$ and $|\mathcal{E}|$ as the number of nodes and edges, respectively, $K$ as the number of environments, $T$ as the number of time slices, $L$ as the number of layers in EA-DGNN, $L'$ as the number of layers in ECVAE, $d$ as the dimension of input node features, $d' = Kd$ as the hidden dimension of EAConv layers in EA-DGNN, $d''$ as the hidden dimension of the encoder and decoder networks layer of ECVAE, $\sum_{v\in\mathcal{V}}\text{Var}(\mathbf{z}_v^{\mathbf{e}'})$ as the variance of $K$ environment-aware representations.

Here we provide a rough analysis of EAGLE's space complexity. Note that, as the space complexity analysis of deep learning models is complicated, we omit some less important terms, such as intermediate activations, *etc.*

- storing the dynamic graph: $\mathcal{O}(KT(|\mathcal{V}| + |\mathcal{E}|))$.
- storing the node input features: $\mathcal{O}(|\mathcal{V}|KTd)$.
- the EAConv layer: $\mathcal{O}(Ld'^2)$.
- the encoder and decoder networks of ECVAE: $\mathcal{O}(L'd'')$.
- storing generated environment samples (the number set to be the same with the observed ones): $\mathcal{O}(|\mathcal{V}|KTd'')$.
- storing the states for function $\mathbb{I}(\cdot,\cdot)$: $\mathcal{O}(K\sum_{v\in\mathcal{V}}\text{Var}(\mathbf{z}_v^{\mathbf{e}'}))$.

Then the overall space complexity of EAGLE can be roughly calculated as:

$$\mathcal{O}(KT(|\mathcal{V}| + |\mathcal{E}|)) + \mathcal{O}(|\mathcal{V}|KTd) + \mathcal{O}(Ld'^2) + \mathcal{O}(L'd'') + \mathcal{O}(|\mathcal{V}|KTd'') + \mathcal{O}(K\sum_{v\in\mathcal{V}}\text{Var}(\mathbf{z}_v^{\mathbf{e}'})). \tag{B.3}$$

However, it is hard to intuitively draw conclusions about memory requirements from the space complexity analysis. Based on our experiments experience, EAGLE can be trained and tested under the hardware configurations (including memory requirements) listed in Appendix E.3, which is on par with the related works' requirements.

# C Proofs

## C.1 Proof of Proposition 1

**Proposition 1.** Given observed environment samples from the dynamic graph $\mathcal{G}^{1:T}$ denoted as

$$\mathbf{z} = \bigcup_{v \in \mathcal{V}} \bigcup_{k=1}^{K} \bigcup_{t=1}^{T} \{\mathbf{z}_{v,k}^{t}\} \in \mathbb{R}^{(|\mathcal{V}| \times K \times T) \times d'} \overset{\text{def}}{=\!=\!=} \mathcal{S}_{\text{ob}} \tag{C.1}$$

with their corresponding one-hot multi-labels $\mathbf{y}$, the environment variable $\mathbf{e}$ is drawn from the prior distribution $p_\omega(\mathbf{e} \mid \mathbf{y})$ across $T$ time slices, and $\mathbf{z}$ is generated from the distribution $p_\omega(\mathbf{z} \mid \mathbf{y}, \mathbf{e})$. Maximizing the conditional log-likelihood $\log p_\omega(\mathbf{z} \mid \mathbf{y})$ leads to an optimal ECVAE by minimizing:

$$\mathcal{L}_{\text{ECVAE}} = \text{KL}[q_\phi(\mathbf{e} \mid \mathbf{z}, \mathbf{y}) \| p_\omega(\mathbf{e} \mid \mathbf{y})] - \frac{1}{|\mathbf{z}|} \sum_{i=1}^{|\mathbf{z}|} \log p_\omega(\mathbf{z} \mid \mathbf{y}, \mathbf{e}^{(i)}), \tag{C.2}$$

where $\text{KL}[\cdot \| \cdot]$ is the Kullback-Leibler (KL) divergence [37], $|\mathbf{z}|$ is the number of observed environment samples, $\mathbf{e}^{(i)}$ is the $i$-th sampling by the reparameterization trick.

*Proof.* The distribution distance between $q_\phi(\mathbf{e} \mid \mathbf{z}, \mathbf{y})$ and $p_\omega(\mathbf{e} \mid \mathbf{z}, \mathbf{y})$ can be calculated by the KL-divergence:

$$
\begin{aligned}
\text{KL}[q_\phi(\mathbf{e} \mid \mathbf{z}, \mathbf{y}) \| p_\omega(\mathbf{e} \mid \mathbf{z}, \mathbf{y})] &= \int q_\phi(\mathbf{e} \mid \mathbf{z}, \mathbf{y}) \log \frac{q_\phi(\mathbf{e} \mid \mathbf{z}, \mathbf{y})}{p_\omega(\mathbf{e} \mid \mathbf{z}, \mathbf{y})} \, d\phi \\
&= \int q_\phi(\mathbf{e} \mid \mathbf{z}, \mathbf{y}) \log \frac{q_\phi(\mathbf{e} \mid \mathbf{z}, \mathbf{y}) p_\omega(\mathbf{z} \mid \mathbf{y}) p_\omega(\mathbf{y})}{p_\omega(\mathbf{e}, \mathbf{z}, \mathbf{y})} \, d\phi \\
&= \int q_\phi(\mathbf{e} \mid \mathbf{z}, \mathbf{y}) \log q_\phi(\mathbf{e} \mid \mathbf{z}, \mathbf{y}) \, d\phi + \underbrace{\int q_\phi(\mathbf{e} \mid \mathbf{z}, \mathbf{y}) \log p_\omega(\mathbf{z} \mid \mathbf{y}) \, d\phi}_{\log p_\omega(\mathbf{z}|\mathbf{y})} \\
&\quad + \int q_\phi(\mathbf{e} \mid \mathbf{z}, \mathbf{y}) \log p_\omega(\mathbf{y}) \, d\phi - \int q_\phi(\mathbf{e} \mid \mathbf{z}, \mathbf{y}) \log p_\omega(\mathbf{e}, \mathbf{z}, \mathbf{y}) \, d\phi \\
&= \log p_\omega(\mathbf{z} \mid \mathbf{y}) + \int q_\phi(\mathbf{e} \mid \mathbf{z}, \mathbf{y}) \log \frac{q_\phi(\mathbf{e} \mid \mathbf{z}, \mathbf{y})}{p_\omega(\mathbf{z} \mid \mathbf{y}, \mathbf{e}) p_\omega(\mathbf{e} \mid \mathbf{y})} \, d\phi \\
&= \log p_\omega(\mathbf{z} \mid \mathbf{y}) + \mathbb{E}_{q_\phi(\mathbf{e}|\mathbf{z},\mathbf{y})}[\log q_\phi(\mathbf{e} \mid \mathbf{z}, \mathbf{y}) - \log p_\omega(\mathbf{e}, \mathbf{z} \mid \mathbf{y})] \\
&= \log p_\omega(\mathbf{z} \mid \mathbf{y}) - \mathbb{E}_{q_\phi(\mathbf{e}|\mathbf{z},\mathbf{y})}[-\log q_\phi(\mathbf{e} \mid \mathbf{z}, \mathbf{y}) + \log p_\omega(\mathbf{e}, \mathbf{z} \mid \mathbf{y})].
\end{aligned}
\tag{C.3}
$$

Thus the conditional log-likelihood $\log p_\omega(\mathbf{z} \mid \mathbf{y})$ can be rewritten as:

$$\log p_\omega(\mathbf{z} \mid \mathbf{y}) = \text{KL}[q_\phi(\mathbf{e} \mid \mathbf{z}, \mathbf{y}) \| p_\omega(\mathbf{e} \mid \mathbf{z}, \mathbf{y})] + \mathbb{E}_{q_\phi(\mathbf{e}|\mathbf{z},\mathbf{y})}[-\log q_\phi(\mathbf{e} \mid \mathbf{z}, \mathbf{y}) + \log p_\omega(\mathbf{e}, \mathbf{z} \mid \mathbf{y})]. \tag{C.4}$$

Since this KL-divergence is non-negative, we then provide an Evidence Lower Bound (ELBO) for $\log p_\omega(\mathbf{y} \mid \mathbf{z})$:

$$
\begin{aligned}
\log p_\omega(\mathbf{z} \mid \mathbf{y}) &\geq \mathbb{E}_{q_\phi(\mathbf{e}|\mathbf{z},\mathbf{y})}[-\log q_\phi(\mathbf{e} \mid \mathbf{z}, \mathbf{y}) + \log p_\omega(\mathbf{e}, \mathbf{z} \mid \mathbf{y})] \\
&= \mathbb{E}_{q_\phi(\mathbf{e}|\mathbf{z},\mathbf{y})}[-\log q_\phi(\mathbf{e} \mid \mathbf{z}, \mathbf{y}) + \log p_\omega(\mathbf{e} \mid \mathbf{y})] + \mathbb{E}_{q_\phi(\mathbf{e}|\mathbf{z},\mathbf{y})}[\log p_\omega(\mathbf{z} \mid \mathbf{y}, \mathbf{e})] \\
&= -\text{KL}[q_\phi(\mathbf{e} \mid \mathbf{z}, \mathbf{y}) \| p_\omega(\mathbf{e} \mid \mathbf{y})] + \mathbb{E}_{q_\phi(\mathbf{e}|\mathbf{z},\mathbf{y})}[\log p_\omega(\mathbf{z} \mid \mathbf{y}, \mathbf{e})].
\end{aligned}
\tag{C.5}
$$

We can maximize $\log p_\omega(\mathbf{z} \mid \mathbf{y})$ by maximizing the ELBO, or minimizing:

$$\mathcal{L}_{\text{ECVAE}} = -\text{ELBO} = \text{KL}[q_\phi(\mathbf{e} \mid \mathbf{z}, \mathbf{y}) \| p_\omega(\mathbf{e} \mid \mathbf{y})] - \mathbb{E}_{q_\phi(\mathbf{e}|\mathbf{z},\mathbf{y})}[\log p_\omega(\mathbf{z} \mid \mathbf{y}, \mathbf{e})]. \tag{C.6}$$

While the second term in $\mathcal{L}_{\text{ECVAE}}$ is the maximum likelihood estimation, which is infeasible to calculate directly under the expectation of the latent environment variable $\mathbf{e} \sim p_\phi(\mathbf{e} \mid \mathbf{z}, \mathbf{y})$ across $T$ time slices. Inspired by Markov Chain Monte Carlo (MCMC) sampling [23], it can be estimated as:

$$\mathcal{L}_{\text{ECVAE}} = \text{KL}[q_\phi(\mathbf{e} \mid \mathbf{z}, \mathbf{y}) \| p_\omega(\mathbf{e} \mid \mathbf{y})] - \frac{1}{|\mathbf{z}|} \sum_{i=1}^{|\mathbf{z}|} \log p_\omega(\mathbf{z} \mid \mathbf{y}, \mathbf{e}^{(i)}). \tag{C.7}$$

In implementations, we assume $q_\phi(\mathbf{e} \mid \mathbf{z}, \mathbf{y})$ and $p_\omega(\mathbf{e} \mid \mathbf{y})$ follow the multivariate normal distribution $\mathcal{N}(\boldsymbol{\mu}; \boldsymbol{\sigma})$ parameterized by $\boldsymbol{\mu}$ and $\boldsymbol{\sigma}$, so that the KL-divergence can be easily calculated. In order to optimize the ECVAE by back-propagation, we utilize a reparameterization trick: $\mathbf{e}^{(i)} = e_\omega(\mathbf{z}, \mathbf{y}, \epsilon^{(i)})$, where $\epsilon^{(i)} \sim \mathcal{N}(\mathbf{0}; \mathbf{I})$. Here $e_\omega(\cdot, \cdot, \cdot)$ is some vector-valued functions parameterized by $\phi$. $\qquad\square$

## C.2 Proof of Proposition 2

**Proposition 2** (A Solution for $\mathbb{I}^\star(\cdot)$). Denote $\mathbf{z}_v^{\mathbf{e}'} = [\mathbf{z}'_{v,1}, \mathbf{z}'_{v,2}, \cdots, \mathbf{z}'_{v,K}]$, where $\mathbf{z}'_{v,k} = \bigcup_{t=1}^{T} \mathbf{z}_{v,k}^t$. Let $\mathrm{Var}(\mathbf{z}_v^{\mathbf{e}'}) \in \mathbb{R}^K$ represents the variance of $K$ environment-aware representations. The Boolean function $\mathbb{I}(\cdot)$ is a solution for $\mathbb{I}^\star(\cdot)$ with the following state update equation:

$$\mathbb{I}(i,j) = \begin{cases} \mathbb{I}(i-1,j) \vee \mathbb{I}(i-1, j - \mathrm{Var}(\mathbf{z}_v^{\mathbf{e}'})[i-1]), & j \geq \mathrm{Var}(\mathbf{z}_v^{\mathbf{e}'})[i-1] \\ \mathbb{I}(i-1,j), & \text{otherwise} \end{cases}, \qquad (\text{C.8})$$

$\mathbb{I}(i,j)$ indicates whether it is feasible to select from the first $i$ elements in $\mathrm{Var}(\mathbf{z}_v^{\mathbf{e}'})$ so that their sum is $j$. Traversing $j$ in reverse order from $\lfloor \sum \mathrm{Var}(\mathbf{z}_v^{\mathbf{e}'})/2 \rfloor$ until satisfying $\mathbb{I}(K, j)$ is True, we reach:

$$\delta_v = \sum \mathrm{Var}(\mathbf{z}_v^{\mathbf{e}'}) - 2j, \qquad (\text{C.9})$$

$$\mathcal{P}_{\mathbf{e}}^I(v) = \left\{ \mathbf{e}_k \mid \mathrm{Var}(\mathbf{z}_v^{\mathbf{e}'})[k] \leq \frac{1}{K} \sum \mathrm{Var}(\mathbf{z}_v^{\mathbf{e}'}) - \frac{\delta_v}{2} \right\}, \quad \mathcal{P}_{\mathbf{e}}^V(v) = \overline{\mathcal{P}_{\mathbf{e}}^I(v)}, \qquad (\text{C.10})$$

where $\delta_v$ is the optimal spatio-temporal invariance threshold of node $v$.

*Proof.* In order to prove the boolean function $\mathbb{I}(\cdot)$ is a solution for $\mathbb{I}^\star(\cdot)$ in Assumption 1, the problem $\mathbb{I}(\cdot)$ solves should satisfies: **(a) Optimal substructure; (b) Non-aftereffect property; (c) Overlapping sub-problems**.

Next, we prove the above three conditions are valid.

**(a) Optimal substructure.** It is said that the problem has the optimal substructure property when the optimal solution of the problem covers the optimal solutions of its subproblems. Now we prove the optimal solution of determining the spatio-temporal invariant patterns $\mathcal{P}_{\mathbf{e}}^I(v)$ for node $v$ is constructed from the optimal solutions of the sub-problems with the bottom-up approach by using the optimal substructure property of the problem.

As $\mathbb{I}(i,j)$ indicates whether it is feasible to select some elements from the first $i$ elements in $\mathrm{Var}(\mathbf{z}_v^{\mathbf{e}'})$ so that their sum is $j$, $\sum \mathrm{Var}(\mathbf{z}_v^{\mathbf{e}'})$ means the sum of elements in $\mathrm{Var}(\mathbf{z}_v^{\mathbf{e}'})$, the target of the problem is to find $\delta_v$ that denotes the optimal threshold for invariance of node $v$. We reduce this problem by dividing $\mathrm{Var}(\mathbf{z}_v^{\mathbf{e}'})$ into two subsets $\mathrm{Var}(\mathbf{z}_v^{\mathbf{e}'})_I$ and $\mathrm{Var}(\mathbf{z}_v^{\mathbf{e}'})_V$, where $\delta_v$ represents the value when the difference between the sum of two subsets is the smallest, *i.e.*,

$$\delta_v = \min\left(\sum \mathrm{Var}(\mathbf{z}_v^{\mathbf{e}'}) - 2j\right), \quad \text{where} \quad \mathbb{I}(K, j) == \text{True}. \qquad (\text{C.11})$$

Let $\mathbb{I}_I(i,j)$ and $\mathbb{I}_V(i,j)$ represent whether it is feasible to select some elements from the first $i$ elements in $\mathrm{Var}(\mathbf{z}_v^{\mathbf{e}'})_I$ and $\mathrm{Var}(\mathbf{z}_v^{\mathbf{e}'})_V$ respectively, so that their sum is $j$. Suppose the sum of $\mathrm{Var}(\mathbf{z}_v^{\mathbf{e}'})_I$ and $\mathrm{Var}(\mathbf{z}_v^{\mathbf{e}'})_V$ are $\sum(\mathrm{Var}(\mathbf{z}_v^{\mathbf{e}'})_I)$ and $\sum(\mathrm{Var}(\mathbf{z}_v^{\mathbf{e}'})_V)$, we have:

$$\sum(\mathrm{Var}(\mathbf{z}_v^{\mathbf{e}'})_I) + \sum(\mathrm{Var}(\mathbf{z}_v^{\mathbf{e}'})_V) = \sum \mathrm{Var}(\mathbf{z}_v^{\mathbf{e}'}). \qquad (\text{C.12})$$

Then, the target of the original problem is transformed into finding an optimal solution for $i$ and $j$, satisfying:

$$\begin{cases} \mathbb{I}_I(i,j) == \text{True}, \\ \mathbb{I}_V(K - i, \sum(\mathrm{Var}(\mathbf{z}_v^{\mathbf{e}'})_I) - j) == \text{True}, \end{cases} \qquad (\text{C.13})$$

$$\textit{s.t.} \ \min\left(\sum(\mathrm{Var}(\mathbf{z}_v^{\mathbf{e}'})_I) - j - \left(\sum(\mathrm{Var}(\mathbf{z}_v^{\mathbf{e}'})_V) - \left(\sum(\mathrm{Var}(\mathbf{z}_v^{\mathbf{e}'})_I) - j\right)\right)\right). \qquad (\text{C.14})$$

According to the definition of $\mathbb{I}(i,j)$, we can draw a conclusion the $\mathbb{I}_I(i,j)$ and $\mathbb{I}_V(i,j)$ both are the subproblems of the original problem, which similarly target at figuring out whether it is feasible to select some elements from the first $i$ elements that satisfy their sum is $j$. So, function $\mathbb{I}_I(i,j)$ and

$\mathbb{I}_V(i, j)$ is the same with $\mathbb{I}(i, j)$ in Eq. (C.8). Assuming that when the difference in the subsets is maximized, $\mathbb{I}_I(i, j)$ and $\mathbb{I}_V(i, j)$ return $j_I$ and $j_V$, respectively, that satisfy:

$$\begin{cases} \mathbb{I}_I(i, j_I) == \text{True}, \\ \mathbb{I}_V(K - i, j_V) == \text{True}. \end{cases} \tag{C.15}$$

We have:

$$\begin{cases} \sum(\text{Var}(\mathbf{z}_v^{\mathbf{e}'})_I) - j_I = \delta_I, & \mathbb{I}_I(i, j_I) == \text{True}, \\ \sum(\text{Var}(\mathbf{z}_v^{\mathbf{e}'})_V) - j_V = \delta_V, & \mathbb{I}_V(K - i, \sum(\text{Var}(\mathbf{z}_v^{\mathbf{e}'})_I) - j_I) == \text{True}. \end{cases} \tag{C.16}$$

As $\mathbb{I}_I(i, j_I)$ and $\mathbb{I}_V(K - i, \sum(\text{Var}(\mathbf{z}_v^{\mathbf{e}'})_I) - j_I)$ are both the optimal solutions, so we reach:

$$|\delta_I - \delta_V| = \left| \sum(\text{Var}(\mathbf{z}_v^{\mathbf{e}'})_I) - j_I - \left( \sum(\text{Var}(\mathbf{z}_v^{\mathbf{e}'})_V) - j_V \right) \right| \leq \delta_v. \tag{C.17}$$

The optimal substructure property is proven.

**(b) Non-aftereffect property.** This property implies that once the state of a certain stage is determined by $\mathbb{I}(\cdot)$, it is not affected by future decision-making. In other words, the subsequent process will not affect the previous state, but only be related to the current state. Eq. (C.8) implies that, when $j \geq \text{Var}(\mathbf{z}_v^{\mathbf{e}'})[i - 1]$, the state is decided by previous state $\mathbb{I}(i - 1, j)$ or $\mathbb{I}(i - 1, j - \text{Var}(\mathbf{z}_v^{\mathbf{e}'})[i - 1])$; accordingly, when $j < \text{Var}(\mathbf{z}_v^{\mathbf{e}'})[i - 1]$, the state is solely decided by previous state $\mathbb{I}(i - 1, j)$. All state transition processes are based on historical states and executed in a one-way transition mode, meeting the requirements of the non-aftereffect property. Generally speaking, the non-aftereffect property is a relaxation condition, that is, as long as the properties of the optimal substructure property are satisfied, the non-aftereffect property will be basically satisfied.

**(c) Overlapping sub-problems.** $\mathbb{I}(\cdot)$ solve the problem from top to bottom in a recursive way, each sub-problem is not always a new problem, but a large number of repeated sub-problems, that is, when different decision sequences reach a certain stage, they will generate duplicate problems. When figuring out $\mathbb{I}(\cdot)$, we consider two situations:

- Not select the $i$-th element in $\text{Var}(\mathbf{z}_v^{\mathbf{e}'})$, *i.e.*, $\mathbb{I}(i, j) = \mathbb{I}(i - 1, j)$;
- Select the $i$-th element in $\text{Var}(\mathbf{z}_v^{\mathbf{e}'})$, *i.e.*, $\mathbb{I}(i, j) = \mathbb{I}(i - 1, j - \text{Var}(\mathbf{z}_v^{\mathbf{e}'})[i - 1])$.

As we can observe, we need the solution of the sub-problem $\mathbb{I}(i-1, j)$ and $\mathbb{I}(i-1, j - \text{Var}(\mathbf{z}_v^{\mathbf{e}'})[i-1])$ when figuring out $\mathbb{I}(i, j)$, which are already calculated by $\mathbb{I}(i-1, \cdot)$. So we can prove that the problem $\mathbb{I}(\cdot)$ solved has overlapping sub-problems.

We then have proven the function $\mathbb{I}(\cdot)$ is a solution for the function $\mathbb{I}^\star(\cdot)$ in Assumption 1, from which we can obtain the optimal spatio-temporal invariance threshold $\delta_v$ of node $v$. Then the spatio-temporal invariant/variant patterns for node $v$ can be exploited by Eq. (C.10), and their unions constitute the overall $\mathcal{P}_{\mathbf{e}}^I$ and $\mathcal{P}_{\mathbf{e}}^V$. We conclude the proof for Proposition 2. $\qquad\square$

### C.3 Proof of Proposition 3

**Proposition 3** (Achievable Assumption). Minimizing Eq. (12) can encourage the model to satisfy the Invariance Property and Sufficient Condition in Assumption 1.

*Proof.* We first propose the following lemma to rewrite the Sufficient Condition and the Invariance Property in Assumption 1 using the information theory [43].

**Lemma 1** (Mutual Information Equivalence). The Invariance Property and Sufficient Condition in Assumption 1 can be equivalently represented with the Mutual Information $I(\cdot; \cdot)$:

**(a) Invariance Property:** $p(\mathbf{Y}^T \mid \mathcal{P}_{\mathbf{e}}^I, \mathbf{e}) = p(\mathbf{Y}^T \mid \mathcal{P}_{\mathbf{e}}^I) \Leftrightarrow I(\mathbf{Y}^T; \mathbf{e} \mid \mathcal{P}_{\mathbf{e}}^I) = 0$;

**(b) Sufficient Condition:** $\mathbf{Y}^T \perp\!\!\!\perp \mathcal{P}_{\mathbf{e}}^V \mid \mathcal{P}_{\mathbf{e}}^I \Leftrightarrow I(\mathbf{Y}^T; \mathcal{P}_{\mathbf{e}}^I)$ is maximized.

*Proof.* We prove Lemma 1 by respectively proving the above two conditions are valid.

**(a) Invariance Property.** According to the definition of the Mutual Information, we can easily get the following equation:

$$I(\mathbf{Y}^T; \mathbf{e} \mid \mathcal{P}_{\mathbf{e}}^I) = \mathrm{KL}\left[p(\mathbf{Y}^T \mid \mathbf{e}, \mathcal{P}_{\mathbf{e}}^I) \parallel p(\mathbf{Y}^T \mid \mathcal{P}_{\mathbf{e}}^I)\right] = 0, \tag{C.18}$$

where $\mathrm{KL}[\cdot\|\cdot]$ is the Kullback-Leibler (KL) divergence [37]. Thus we have proved the equivalence in Invariance Property.

**(b) Sufficient Condition.** We demonstrate sufficiency and necessity through the following two steps.

**First**, we prove that for $\mathbf{Y}^T$, $\mathcal{P}_{\mathbf{e}}^I$ and $\epsilon$ satisfying $\mathbf{Y}^T = g(\mathbf{Z}_I^{1:T}) + \epsilon$ would also satisfy $\mathcal{P}_{\mathbf{e}}^I = \arg\max_{\mathcal{P}_{\mathbf{e}}^I} I(\mathbf{Y}^T; \mathcal{P}_{\mathbf{e}}^I)$. We perform proving by contradiction. Suppose $\mathcal{P}_{\mathbf{e}}^I \neq \arg\max_{\mathcal{P}_{\mathbf{e}}^I} I(\mathbf{Y}^T; \mathcal{P}_{\mathbf{e}}^I)$ and there exists $\mathcal{P}_{\mathbf{e}}^{I'} = \arg\max_{\mathcal{P}_{\mathbf{e}}^I}$ where $\mathcal{P}_{\mathbf{e}}^{I'} \neq \mathcal{P}_{\mathbf{e}}^I$. We can always find a mapping function $\mathcal{M}$ so that $\mathcal{P}_{\mathbf{e}}^{I'} = \mathcal{M}(\mathcal{P}_{\mathbf{e}}^I, \pi)$ where $\pi$ is a random variable. Then we reach:

$$I(\mathbf{Y}^T; \mathcal{P}_{\mathbf{e}}^{I'}) = I(\mathbf{Y}^T; \mathcal{P}_{\mathbf{e}}^I, \pi) = I(g(\mathbf{Z}_I^{1:T}); \mathcal{P}_{\mathbf{e}}^I, \pi) = I(g(\mathbf{Z}_I^{1:T}); \mathcal{P}_{\mathbf{e}}^I) = I(\mathbf{Y}^T; \mathcal{P}_{\mathbf{e}}^I), \tag{C.19}$$

which leads to a contradiction.

**Next**, we prove that for $\mathbf{Y}^T$, $\mathcal{P}_{\mathbf{e}}^I$ and $\epsilon$ satisfying $\mathcal{P}_{\mathbf{e}}^I = \arg\max_{\mathcal{P}_{\mathbf{e}}^I} I(\mathbf{Y}^T; \mathcal{P}_{\mathbf{e}}^I)$ would also satisfy $\mathbf{Y}^T = g(\mathbf{Z}_I^{1:T}) + \epsilon$. We also perform proving by contradiction. Suppose that $\mathbf{Y}^T \neq g(\mathbf{Z}_I^{1:T}) + \epsilon$ and $\mathbf{Y}^T = g(\mathbf{Z}_{I'}^{1:T}) + \epsilon$ where $\mathcal{P}_{\mathbf{e}}^{I'} \neq \mathcal{P}_{\mathbf{e}}^I$. We then have the following inequality:

$$I(g(\mathbf{Z}_{I'}^{1:T}); \mathcal{P}_{\mathbf{e}}^I) \leq I(g(\mathbf{Z}_{I'}^{1:T}); \mathcal{P}_{\mathbf{e}}^{I'}), \tag{C.20}$$

from which we can obtain that $\mathcal{P}_{\mathbf{e}}^{I'} = \arg\max_{\mathcal{P}_{\mathbf{e}}^I} I(\mathbf{Y}^T; \mathcal{P}_{\mathbf{e}}^I)$, leading to a contradiction. $\square$

Now, we prove Proposition 3 in the following two perspectives.

**First**, we prove that minimizing the expectation term ($\mathcal{L}_{\text{task}}$) in Eq. (12) can enforce the model to satisfy the Sufficient Condition in Assumption 1 (note that we omit the subscript $\mathcal{P}_{\mathbf{e}}^I$ in $\mathbf{Z}_I^{1:T}$ for brevity in the rest of this subsection if there are no special declarations).

From the SCM model in Figure 1(b), we can analyze that $\max_{q(\mathbf{Z}^{1:T}|\mathbf{G}^{1:T})} I(\mathbf{Y}^T; \mathbf{Z}^{1:T})$ is equivalent to $\min_{q(\mathbf{Z}^{1:T}|\mathbf{G}^{1:T})} I(\mathbf{Y}^T; \mathbf{G}^{1:T} \mid \mathbf{Z}^{1:T})$, as we filter out the spurious correlations that contain in the dependencies between $\mathbf{Y}^T$ and $\mathbf{G}^{1:T}$. Treating $q(\mathbf{Z}^{1:T} \mid \mathbf{G}^{1:T})$ as the variational distribution, we have the following upper bound:

$$\begin{aligned}
I(\mathbf{Y}^T; \mathbf{G}^{1:T} \mid \mathbf{Z}^{1:T}) &= \mathrm{KL}\left[p(\mathbf{Y}^T \mid \mathbf{G}^{1:T}, \mathbf{e}) \parallel p(\mathbf{Y}^T \mid \mathbf{Z}^{1:T}, \mathbf{e})\right] \\
&= \mathrm{KL}\left[p(\mathbf{Y}^T \mid \mathbf{G}^{1:T}, \mathbf{e}) \parallel q(\mathbf{Y}^T \mid \mathbf{Z}^{1:T})\right] \\
&\quad - \mathrm{KL}\left[p(\mathbf{Y}^T \mid \mathbf{Z}^{1:T}, \mathbf{e}) \parallel q(\mathbf{Y}^T \mid \mathbf{Z}^{1:T})\right] \\
&\leq \mathrm{KL}\left[p(\mathbf{Y}^T \mid \mathbf{G}^{1:T}, \mathbf{e}) \parallel q(\mathbf{Y}^T \mid \mathbf{Z}^{1:T})\right] \\
&\leq \min_{q(\mathbf{Y}^T|\mathbf{Z}^{1:T})} \mathrm{KL}\left[p(\mathbf{Y}^T \mid \mathbf{G}^{1:T}, \mathbf{e}) \parallel q(\mathbf{Y}^T \mid \mathbf{Z}^{1:T})\right].
\end{aligned} \tag{C.21}$$

In addition, we have:

$$\begin{aligned}
&\mathrm{KL}\left[p(\mathbf{Y}^T \mid \mathbf{G}^{1:T}, \mathbf{e}) \parallel q(\mathbf{Y}^T \mid \mathbf{Z}^{1:T})\right] \\
&= \mathbb{E}_{\mathbf{e}} \mathbb{E}_{(\mathcal{G}^{1:T}, Y^T) \sim p(\mathbf{G}^{1:T}, \mathbf{Y}^T|\mathbf{e})} \mathbb{E}_{\mathbf{Z}^{1:T} \sim q(\mathbf{Z}^{1:T}|\mathbf{G}^{1:T}=\mathcal{G}^{1:T})} \left[\log \frac{p(\mathbf{Y}^T = Y^T \mid \mathbf{G}^{1:T} = \mathcal{G}^{1:T}, \mathbf{e})}{q(\mathbf{Y}^T = Y^T \mid \mathbf{Z}^{1:T})}\right] \\
&\leq \mathbb{E}_{\mathbf{e}} \mathbb{E}_{(\mathcal{G}^{1:T}, Y^T) \sim p(\mathbf{G}^{1:T}, \mathbf{Y}^T|\mathbf{e})} \left[\log \frac{(\mathbf{Y}^T = Y^T \mid \mathbf{G}^{1:T} = \mathcal{G}^{1:T}, \mathbf{e})}{\mathbb{E}_{\mathbf{Z}^{1:T} \sim q(\mathbf{Z}^{1:T}|\mathbf{G}^{1:T}=\mathcal{G}^{1:T})}[q(\mathbf{Y}^T = Y^T \mid \mathbf{Z}^{1:T})]}\right].
\end{aligned} \tag{C.22}$$

The inequality in Eq. (C.22) is derived from Jensen's Inequality [34] and [86], while the EA-DGNN $w(\cdot)$ ensures $q(\mathbf{Z}^{1:T} \mid \mathbf{G}^{1:T})$ is a Dirac delta distribution ($\delta$-distribution). Then we reach:

$$\begin{aligned}
&\min_{q(\mathbf{Y}^T|\mathbf{Z}^{1:T})} \mathrm{KL}\left[p(\mathbf{Y}^T \mid \mathbf{G}^{1:T}, \mathbf{e}) \parallel q(\mathbf{Y}^T \mid \mathbf{Z}^{1:T})\right] \\
&\Leftrightarrow \min_{\boldsymbol{\theta}} \mathbb{E}_{\mathbf{e} \sim q_\phi(\mathbf{e}), (\mathcal{G}^{1:T}, Y^T) \sim p(\mathbf{G}^{1:T}, \mathbf{Y}^T|\mathbf{e})} \left[\ell\left(g(\mathbf{Z}_I^{1:T}), Y^T\right)\right].
\end{aligned} \tag{C.23}$$

We thus have proven that, minimizing the expectation term ($\mathcal{L}_{\text{task}}$) in Eq. (12) is to minimize the upper bound of $I(\mathbf{Y}^T; \mathbf{G}^{1:T} \mid \mathbf{Z}^{1:T})$ (maximize the lower bound of $\max_{q(\mathbf{Z}^{1:T}|\mathbf{G}^{1:T})} I(\mathbf{Y}^T; \mathbf{Z}^{1:T})$), which leads to maximizing $I(\mathbf{Y}^T; \mathcal{P}_{\mathbf{e}}^I)$. This helps enforce the model to satisfy the Sufficient Condition.

**Then**, we prove that minimizing the variance term ($\mathcal{L}_{\text{risk}}$) in Eq. (12) can enforce the model to satisfy the Invariance Property in Assumption 1.

We have:

$$
\begin{aligned}
I(\mathbf{Y}^T; \mathbf{e} \mid \mathbf{Z}^{1:T}) &= \mathrm{KL}\left[ p(\mathbf{Y}^T \mid \mathbf{Z}^{1:T}, \mathbf{e}) \parallel p(\mathbf{Y}^T \mid \mathbf{Z}^{1:T}) \right] \\
&= \mathrm{KL}\left[ p(\mathbf{Y}^T \mid \mathbf{Z}^{1:T}, \mathbf{e}) \parallel \mathbb{E}_{\mathbf{e}}\left[ p(\mathbf{Y}^T \mid \mathbf{Z}^{1:T}, \mathbf{e}) \right] \right] \\
&= \mathrm{KL}\left[ q(\mathbf{Y}^T \mid \mathbf{Z}^{1:T}) \parallel \mathbb{E}_{\mathbf{e}} q(\mathbf{Y}^T \mid \mathbf{Z}^{1:T}) \right] \\
&\quad - \mathrm{KL}\left[ q(\mathbf{Y}^T \mid \mathbf{Z}^{1:T}) \parallel p(\mathbf{Y}^T \mid \mathbf{Z}^{1:T}, \mathbf{e}) \right] \\
&\quad - \mathrm{KL}\left[ \mathbb{E}_{\mathbf{e}}\left[ p(\mathbf{Y}^T \mid \mathbf{Z}^{1:T}, \mathbf{e}) \right] \parallel \mathbb{E}_{\mathbf{e}}\left[ q(\mathbf{Y}^T \mid \mathbf{Z}^{1:T}) \right] \right] \\
&\leq \mathrm{KL}\left[ q(\mathbf{Y}^T \mid \mathbf{Z}^{1:T}) \parallel \mathbb{E}_{\mathbf{e}} q(\mathbf{Y}^T \mid \mathbf{Z}^{1:T}) \right] \\
&\leq \min_{q(\mathbf{Y}^T|\mathbf{Z}^{1:T})} \mathrm{KL}\left[ q(\mathbf{Y}^T \mid \mathbf{Z}^{1:T}) \parallel \mathbb{E}_{\mathbf{e}} q(\mathbf{Y}^T \mid \mathbf{Z}^{1:T}) \right].
\end{aligned}
\tag{C.24}
$$

In addition, we have:

$$
\begin{aligned}
&\mathrm{KL}\left[ q(\mathbf{Y}^T \mid \mathbf{Z}^{1:T}) \parallel \mathbb{E}_{\mathbf{e}} q(\mathbf{Y}^T \mid \mathbf{Z}^{1:T}) \right] \\
&= \mathbb{E}_{\mathbf{e}} \mathbb{E}_{(\mathcal{G}^{1:T}, Y^T) \sim p(\mathbf{G}^{1:T}, \mathbf{Y}^T | \mathbf{e})} \mathbb{E}_{\mathbf{Z}^{1:T} \sim q(\mathbf{Z}^{1:T} | \mathbf{G}^{1:T} = \mathcal{G}^{1:T})} \left[ \log \frac{q(\mathbf{Y}^T = Y^T \mid \mathbf{Z}^{1:T})}{\mathbb{E}_{\mathbf{e}} q(\mathbf{Y}^T = Y^T \mid \mathbf{Z}^{1:T})} \right].
\end{aligned}
\tag{C.25}
$$

Derived from Jensen's Inequality, the upper bound for $\mathrm{KL}\left[ q(\mathbf{Y}^T \mid \mathbf{Z}^{1:T}) \parallel \mathbb{E}_{\mathbf{e}} q(\mathbf{Y}^T \mid \mathbf{Z}^{1:T}) \right]$ is:

$$
\begin{aligned}
&\mathrm{KL}\left[ q(\mathbf{Y}^T \mid \mathbf{Z}^{1:T}) \parallel \mathbb{E}_{\mathbf{e}} q(\mathbf{Y}^T \mid \mathbf{Z}^{1:T}) \right] \\
&\leq \mathbb{E}_{\mathbf{e}}\left[ \left| \ell\left( f_{\boldsymbol{\theta}}\left( \mathcal{G}^{1:T} \right), Y^T \right) - \mathbb{E}_{\mathbf{e}}\left[ \ell\left( f_{\boldsymbol{\theta}}\left( \mathcal{G}^{1:T} \right), Y^T \right) \right] \right| \right].
\end{aligned}
\tag{C.26}
$$

Finally, we reach:

$$
\begin{aligned}
&\min_{q(\mathbf{Y}^T|\mathbf{Z}^{1:T})} \mathrm{KL}\left[ q(\mathbf{Y}^T \mid \mathbf{Z}^{1:T}) \parallel \mathbb{E}_{\mathbf{e}} q(\mathbf{Y}^T \mid \mathbf{Z}^{1:T}) \right] \\
&\Leftrightarrow \min_{\boldsymbol{\theta}} \mathrm{Var}_{s \in \mathcal{S}} \left\{ \mathbb{E}_{\mathbf{e} \sim q_{\phi}(\mathbf{e}), (\mathcal{G}^{1:T}, Y^T) \sim p(\mathbf{G}^{1:T}, \mathbf{Y}^T | \mathbf{e})} \left[ \ell\left( f_{\boldsymbol{\theta}}\left( \mathcal{G}^{1:T} \right), Y^T \mid \mathrm{do}(\mathbf{Z}_V^{1:T} = s) \right) \right] \right\}.
\end{aligned}
\tag{C.27}
$$

We thus have proven that, minimizing the variance term ($\mathcal{L}_{\text{risk}}$) in Eq. (12) is to minimize the upper bound of $I(\mathbf{Y}^T; \mathbf{e} \mid \mathbf{Z}^{1:T})$, which leads to minimizing $I(\mathbf{Y}^T; \mathbf{e} \mid \mathcal{P}_{\mathbf{e}}^I)$. This helps enforce the model to satisfy the Invariance Property.

We conclude the proof for Proposition 3. $\qquad\square$

### C.4  Proof of Proposition 4

**Proposition 4** (Equivalent Optimization). *Optimizing Eq. (12) is equivalent to minimizing the upper bound of the OOD generalization error in Eq. (2).*

*Proof.* As we defined in Eq. (2), the generation of dynamic graph data $(\mathcal{G}^{1:T}, Y^T)$ is drawn from the distribution $p(\mathbf{G}^{1:T}, \mathbf{Y}^T \mid \mathbf{e})$, while the difference of $\mathbf{e}$ during training and testing causes the out-of-distribution shifts. Let $q(\mathbf{Y}^T \mid \mathbf{G}^{1:T})$ be the inferred variational distribution of the ground-truth distribution $p(\mathbf{Y}^T \mid \mathbf{G}^{1:T}, \mathbf{e})$, then the OOD generalization error can be measured by the KL-divergence of the two distributions:

$$
\begin{aligned}
&\mathrm{KL}\left[ p(\mathbf{Y}^T \mid \mathbf{G}^{1:T}, \mathbf{e}) \parallel q(\mathbf{Y}^T \mid \mathbf{G}^{1:T}) \right] \\
&= \mathbb{E}_{\mathbf{e}} \mathbb{E}_{(\mathcal{G}^{1:T}, Y^T) \sim p(\mathbf{G}^{1:T}, \mathbf{Y}^T | \mathbf{e})} \left[ \log \frac{p(\mathbf{Y}^T = Y^T \mid \mathbf{G}^{1:T} = \mathcal{G}^{1:T}, \mathbf{e})}{q(\mathbf{Y}^T = Y^T \mid \mathbf{G}^{1:T} = \mathcal{G}^{1:T})} \right].
\end{aligned}
\tag{C.28}
$$

Inspired by [19, 86], we apply an information-theoretic approach to our scenarios. First, we propose the following lemma in order to rewrite the OOD generalization error.

**Lemma 2** (OOD Generalization Upper Bound). The OOD generalization error is upper bounded by:

$$\mathrm{KL}\left[p(\mathbf{Y}^T \mid \mathbf{G}^{1:T}, \mathbf{e}) \parallel q(\mathbf{Y}^T \mid \mathbf{G}^{1:T})\right] \leq \mathrm{KL}\left[p(\mathbf{Y}^T \mid \mathbf{G}^{1:T}, \mathbf{e}) \parallel q(\mathbf{Y}^T \mid \mathbf{Z}^{1:T})\right], \qquad (C.29)$$

where $q(\mathbf{Y}^T \mid \mathbf{Z}^{1:T})$ can be seen as the inferred variational distribution of the edge predictor.

*Proof.* We prove Lemma 2 by continue investigating on Eq. (C.28):

$$\mathrm{KL}\left[p(\mathbf{Y}^T \mid \mathbf{G}^{1:T}, \mathbf{e}) \parallel q(\mathbf{Y}^T \mid \mathbf{G}^{1:T})\right]$$

$$= \mathbb{E}_{\mathbf{e}}\mathbb{E}_{(\mathcal{G}^{1:T}, Y^T) \sim p(\mathbf{G}^{1:T}, \mathbf{Y}^T \mid \mathbf{e})}\left[\log \frac{p(\mathbf{Y}^T = Y^T \mid \mathbf{G}^{1:T} = \mathcal{G}^{1:T}, \mathbf{e})}{q(\mathbf{Y}^T = Y^T \mid \mathbf{G}^{1:T} = \mathcal{G}^{1:T})}\right]$$

$$= \mathbb{E}_{\mathbf{e}}\mathbb{E}_{(\mathcal{G}^{1:T}, Y^T) \sim p(\mathbf{G}^{1:T}, \mathbf{Y}^T \mid \mathbf{e})}\left[\log \frac{p(\mathbf{Y}^T = Y^T \mid \mathbf{G}^{1:T} = \mathcal{G}^{1:T}, \mathbf{e})}{\mathbb{E}_{\mathbf{Z}^{1:T} \sim q(\mathbf{Z}^{1:T} \mid \mathbf{G}^{1:T} = \mathcal{G}^{1:T})}\left[q(\mathbf{Y}^T = Y^T \mid \mathbf{Z}^{1:T})\right]}\right]$$

$$\leq \mathbb{E}_{\mathbf{e}}\mathbb{E}_{(\mathcal{G}^{1:T}, Y^T) \sim p(\mathbf{G}^{1:T}, \mathbf{Y}^T \mid \mathbf{e})}\mathbb{E}_{\mathbf{Z}^{1:T} \sim q(\mathbf{Z}^{1:T} \mid \mathbf{G}^{1:T} = \mathcal{G}^{1:T})}\left[\log \frac{p(\mathbf{Y}^T = Y^T \mid \mathbf{G}^{1:T} = \mathcal{G}^{1:T}, \mathbf{e})}{q(\mathbf{Y}^T = Y^T \mid \mathbf{Z}^{1:T})}\right]$$

$$= \mathrm{KL}\left[p(\mathbf{Y}^T \mid \mathbf{G}^{1:T}, \mathbf{e}) \parallel q(\mathbf{Y}^T \mid \mathbf{Z}^{1:T})\right]. \quad \text{(upper bound for OOD generalization error)}$$
$$(C.30)$$

Again, the inequality in Eq. (C.22) is derived from Jensen's Inequality, while the EA-DGNN $w(\cdot)$ ensures $q(\mathbf{Z}^{1:T} \mid \mathbf{G}^{1:T})$ is a Dirac delta distribution ($\delta$-distribution). $\square$

Based on Lemma 1, we can adapt the Eq. (12) as:

$$\min_{q(\mathbf{Z}^{1:T} \mid \mathbf{G}^{1:T}), q(\mathbf{Y}^T, \mathbf{Z}^{1:T})} \mathrm{KL}\left[p(\mathbf{Y}^T \mid \mathbf{G}^{1:T}, \mathbf{e}) \parallel q(\mathbf{Y}^T \mid \mathbf{Z}^{1:T})\right] + I(\mathbf{Y}^T; \mathbf{e} \mid \mathbf{Z}^{1:T}). \qquad (C.31)$$

Thus, based on Lemma 2, we validate that minimizing Eq. (12) is equivalent to minimizing the upper bound of the OOD generalization error in Eq. (2), *i.e.*,

$$\min_{\boldsymbol{\theta}} \mathcal{L}_{\mathrm{task}} + \alpha\mathcal{L}_{\mathrm{risk}} \Leftrightarrow \min_{q(\mathbf{Z}^{1:T} \mid \mathbf{G}^{1:T}), q(\mathbf{Y}^T, \mathbf{Z}^{1:T})} \mathrm{KL}\left[p(\mathbf{Y}^T \mid \mathbf{G}^{1:T}, \mathbf{e}) \parallel q(\mathbf{Y}^T \mid \mathbf{Z}^{1:T})\right]$$

$$+ I(\mathbf{Y}^T; \mathbf{e} \mid \mathbf{Z}^{1:T})$$

$$\geq \min_{q(\mathbf{Z}^{1:T} \mid \mathbf{G}^{1:T}), q(\mathbf{Y}^T, \mathbf{Z}^{1:T})} \mathrm{KL}\left[p(\mathbf{Y}^T \mid \mathbf{G}^{1:T}, \mathbf{e}) \parallel q(\mathbf{Y}^T \mid \mathbf{Z}^{1:T})\right]$$

$$\geq \mathrm{KL}\left[p(\mathbf{Y}^T \mid \mathbf{G}^{1:T}, \mathbf{e}) \parallel q(\mathbf{Y}^T \mid \mathbf{G}^{1:T})\right]. \, (I(\mathbf{Y}^T; \mathbf{e} \mid \mathbf{Z}^{1:T}) \text{ is non-negative})$$
$$(C.32)$$

We conclude the proof for Proposition 4. $\square$

## D  Experiment Details and Additional Results

### D.1  Datasets Details

We use three real-world datasets to evaluate EAGLE on the challenging future link prediction task.

- **COLLAB**[1] [81] is an academic collaboration dataset with papers that were published during 1990-2006 (16 graph snapshots). Nodes and edges represent authors and co-authorship, respectively. Based on the co-authored publication, there are five attributes in edges, including "Data Mining", "Database", "Medical Informatics", "Theory" and "Visualization". We pick "Data Mining" as the shifted attribute. We apply word2vec [59] to extract 32-dimensional node features from paper abstracts. We use 10/1/5 chronological graph snapshots for training, validation, and testing, respectively. The dataset includes 23,035 nodes and 151,790 links in total.

- **Yelp**[2] [75] contains customer reviews on business. Nodes and edges represent customer/business and review behaviors, respectively. Considering categories of business, there are five attributes in edges, including "Pizza", "American (New) Food", "Coffee & Tea", "Sushi Bars" and "Fast Food" from January 2019 to December 2020 (24 graph snapshots). We pick "Pizza" as the shifted attribute. We apply word2vec [59] to extract 32-dimensional node features from reviews. We use 15/1/8 chronological graph snapshots for training, validation, and testing, respectively. The dataset includes 13,095 nodes and 65,375 links in total.

---

[1] https://www.aminer.cn/collaboration
[2] https://www.yelp.com/dataset

- **ACT**[3] [45] describes student actions on a MOOC platform within a month (30 graph snapshots). Nodes represent students or targets of actions, edges represent actions. Considering the attributes of different actions, we apply K-Means [29] to cluster the action features into five categories and randomly select a certain category (the 5th cluster) of edges as the shifted attribute. We assign the features of actions to each student or target and expand the original 4-dimensional features to 32 dimensions by a linear function. We use 20/2/8 chronological graph snapshots for training, validation, and testing, respectively. The dataset includes 20,408 nodes and 202,339 links in total.

Statistics of the three datasets are concluded in Table D.1. These three datasets have different time spans and temporal granularity (16 years, 24 months, and 30 days), covering most real-world scenarios. The most challenging dataset for the future link prediction task is the COLLAB. In addition to having the longest time span and the coarsest temporal granularity, it also has the largest difference in the properties of its links.

Table D.1: Statistics of the real-world datasets.

| Dataset | # Nodes | # Links | # Graph Snapshots | Temporal Granularity | In-distribution Attributes | Shifted Attribute |
|---------|---------|---------|-------------------|---------------------|----------------------------|-------------------|
| COLLAB | 23,035 | 151,790 | 16 | year | Database, Medical Informatics, Theory, Visualization | Data Mining |
| Yelp | 13,095 | 65,375 | 24 | month | American (New) Food, Fast Food Sushi Bars, Coffee & Tea | Pizza |
| ACT | 20,408 | 202,339 | 30 | day | Attributes 1-4 | Attribute 5 |

We visualize the distribution shifts in the three real-world dataset with respect to the average neighbor degree (Figure D.1) and the number of interactions (Figure D.2) in training and testing sets. We observe that, there exists a huge difference in terms of the values, trends, *etc.*, between the training set and the testing set, which demonstrates the distribution shifts are heavy. Interestingly, COLLAB has less testing data than its training data, which is common in real-world scenarios, such as not all the co-authorship was established from the beginning. In addition, we notice a drastic drop in Yelp after January 2019 when the COVID-19 outbreak. The sudden change in predictive patterns increases the difficulty of the task. A similar abnormal steep upward trend can also be witnessed in ACT after Day 20, which may be caused by an unknown out-of-distribution event.

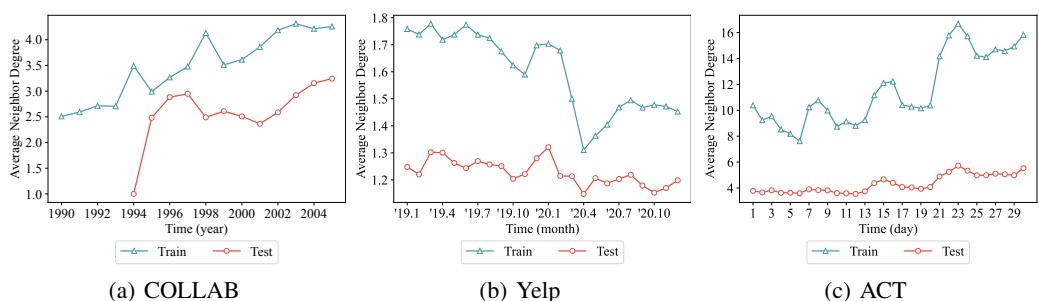

(a) COLLAB        (b) Yelp        (c) ACT

Figure D.1: Visualizations of the average neighbor degree in each graph snapshot.

### D.2 Baseline Details

We compare EAGLE with representative GNNs and OOD generalization methods.

- **Static GNNs**: **GAE** [42] is a representative static GNN as the GCN [41] based graph autoencoder; **VGAE** [42] further introduces variational variables into GAE, possessing better generative ability.
- **Dynamic GNNs**: **GCRN** [76] is a representative dynamic GNN following "spatial first, temporal second" convolution mechanism, which firstly adopts GCNs to obtain node embeddings and then a GRU [13] to capture temporal relations; **EvolveGCN** [62] applies an LSTM [31] or GRU

---

[3]https://snap.stanford.edu/data/act-mooc.html

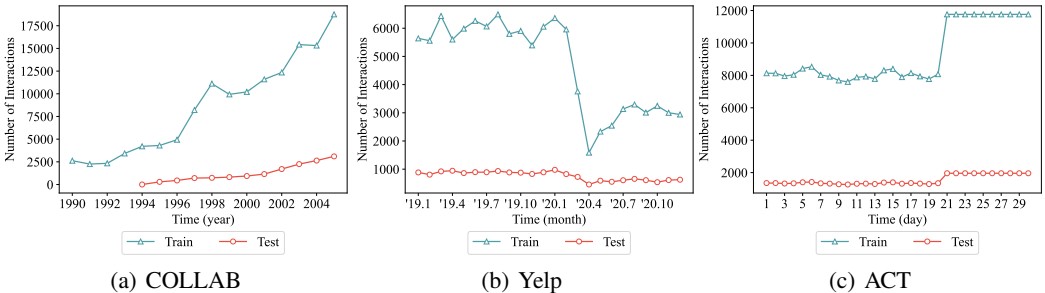

Figure D.2: Visualizations of the number of interactions in each graph snapshot.

to flexibly evolve the parameters of GCNs instead of modeling the dynamics after deriving node embeddings; **DySAT** [75] models dynamic graph through self-attentions in both structural neighborhoods and temporal dynamics.

- **OOD generalization methods**: **IRM** [3] minimizes the empirical risk to learn an optimal invariant predictor under potential environments; **V-REx** [44] extends the IRM by reweighting the empirical risk to emphasize more on training samples with larger errors; **GroupDRO** [74] reduces the empirical risk gap across training distributions to enhance the robustness when encountering heavy OOD shifts; **DIDA** [94] tackles OOD generalization problem on dynamic graphs for the first time by discovering and utilizing invariant patterns. It is worth noting that, DIDA [94] is the most relative work as our main baseline for comparison.

### D.3  Experiment Setting Details

**Detailed Settings for Section 4.1.1.** Each of the three real-world datasets can be split into several partial dynamic graphs based on their link properties, which demonstrates the multi-attribute relations under the impact of their surrounding environments. We filter out one certain attribute links as the variables under the future shifted environment as the OOD data, and the left links are further divided into training, validation, and testing sets chronologically. The shifted attribute links will only be accessible during the OOD testing stage, which is more practical and challenging in real-world scenarios as the model cannot capture any information about the filtered links during training and validation. Note that, all attribute-related features have been removed after the above operations before feeding to EAGLE. Take the COLLAB dataset for example. There are five attribute links in COLLAB as summarized in Table D.1. We filter out all the links with the attribute "Data Mining", and split the rest of the links into training, validation, and testing sets by positive and negative edge sampling. Then we add the "Data Mining" links into testing sets to make the distribution shifts. Finally, we remove all link attribute information to avoid data leakage.

**Detailed Settings for Section 4.1.2.** Denote original node features and structures as $\mathbf{X}^t \in \mathbb{R}^{N \times d}$ and $\mathbf{A}^t \in \{0, 1\}^{N \times N}$. For each time $t$, we uniformly sample $p(t)|\mathcal{E}^{t+1}|$ positive links and $(1 - p(t))|\mathcal{E}^{t+1}|$ negative links, which are then factorized into shifted features $\mathbf{X}^{t\prime} \in \mathbb{R}^{N \times d}$ while preserving structural property. Original node features and synthesized node features are concatenated as $[\mathbf{X}^t \| \mathbf{X}^{t\prime}]$ as the input. In details, $\mathbf{X}^{t\prime}$ is obtained by training the embeddings with reconstruction loss $\ell(\mathbf{X}^{t\prime}\mathbf{X}^{t\prime\top}, \mathbf{A}^{t+1})$, where $\ell(\cdot)$ refers to the cross-entropy loss function [16]. In this way, we find that the link predictor can achieve satisfying results by using $\mathbf{X}^{t\prime}$ to predict the links in $\mathbf{A}^{t+1}$, which demonstrates that the generated node features have strong correlations with the future underlying environments. The sampling probability $p(t) = \bar{p} + \sigma \cos(t)$, where $\mathbf{X}^{t\prime}$ with higher $p(t)$ will have stronger spurious correlations with future underlying environments. Note that, we apply the $\mathrm{clip}(\cdot)$ function to limit the probability to between 0 and 1. We set $\bar{p}$ to be 0.4, 0.6, and 0.8 for training and 0.1 for testing; set $\sigma = 0.05$ in training and $\sigma = 0$ in testing.

**Detailed Settings for Section 4.2.** We set the number of nodes $N = 2,000$ with 10 graph snapshots, where 6/2/2 chronological snapshots are used for training, validation, and testing, respectively. We set $K = 5$ and let $\sigma_{\mathbf{e}}$ represent the proportion of the environments in which the invariant patterns are learned, where higher $\sigma_{\mathbf{e}}$ means more reliable invariant patterns. Node features with respect to different environments are drawn from five multivariate normal distributions $\mathcal{N}(\boldsymbol{\mu}_k; \boldsymbol{\sigma}_k)$. Features

with respect to the invariant patterns will be perturbed slightly, while features with respect to the variant patterns will be perturbed significantly. Here we perturb features by adding Gaussian noise with different degrees. We then construct graph structures based on node feature similarity. Links generated by the node-pair representations under the $\mathbf{e}_5$ are filtered out during the training and validation stages, which is similar to the setting of Section 4.1.1, and they only appear in the testing stage following the proportion constraint $\bar{q}$. Higher $\bar{q}$ means more heavier distribution shifts. $\mathbb{I}_{\mathrm{ACC}}$ denotes the prediction accuracy of the invariant patterns by $\mathbb{I}(\cdot)$. As the environments $\mathbf{e} = \{\mathbf{e}_k\}_{k=1}^5$ do not satisfy the permutation invariance property, thus the predicted invariant patterns with respect to $\mathbf{e}$ is hard to evaluate. May wish to set the $\mathbb{I}_{\mathrm{ACC}}$ reports the highest results as we shift the orders of environments in $\mathbf{e}$ to satisfy the predicted invariant patterns better.

## D.4 Hyperparameter Sensitivity Analysis

We analyze the sensitivity of the hyperparameters $\alpha$ and $\beta$, which act as the trade-off for loss in Eq. (16). The hyperparameter $\alpha$ is chosen from $\{10^{-3}, 10^{-2}, 10^{-1}, 10^0, 10^1\}$, and $\beta$ is chosen from $\{10^{-6}, 10^{-5}, 10^{-4}, 10^{-3}, 10^{-2}\}$. We conduct analysis on three real-world datasets and report results in Figure D.3 and Figure D.4. Results demonstrate that the task performance experiences a significant decline in most datasets when the values of $\alpha$ and $\beta$ are too large or too small. We can draw a conclusion that $\alpha$ acts as a balance factor between exploiting the spatio-temporal invariant patterns for out-of-distribution prediction and generalizing to diverse latent environments with respect to variant patterns. $\beta$ plays a role in balancing the trade-off between modeling the environment and inferring the environment distribution as a bi-level optimization. In conclusion, different combinations of hyperparameters lead to varying task performance, and we follow the tradition of reporting the best task performance with standard deviations.

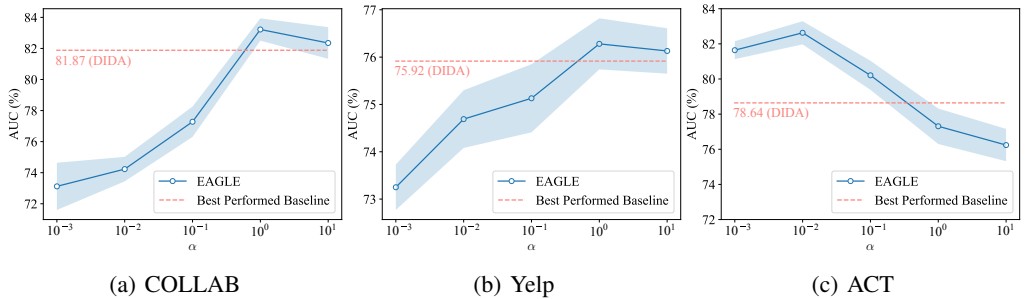

|       |       |       |
| :---: | :---: | :---: |
| (a) COLLAB | (b) Yelp | (c) ACT |

Figure D.3: Sensitivity analysis of the hyperparameter $\alpha$ on three real-world datasets. The solid line shows the average AUC (%) in the testing stage and the light blue area represents standard deviations. The dashed line represents the average AUC (%) of the best-performed baseline.

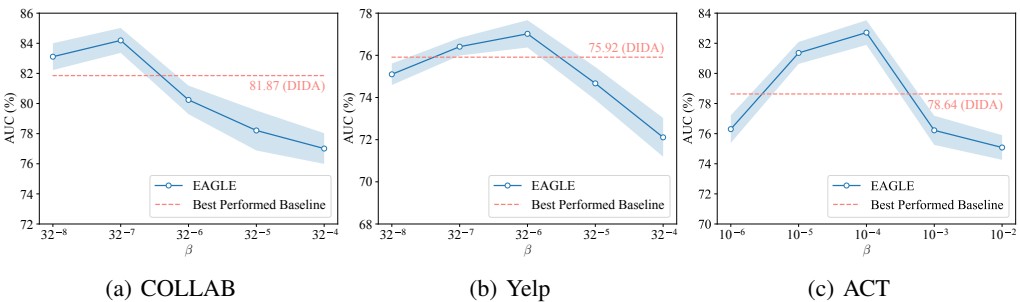

|       |       |       |
| :---: | :---: | :---: |
| (a) COLLAB | (b) Yelp | (c) ACT |

Figure D.4: Sensitivity analysis of the hyperparameter $\beta$ on three real-world datasets. The solid line shows the average AUC (%) in the testing stage and the light blue area represents standard deviations. The dashed line represents the average AUC (%) of the best-performed baseline.

## D.5 Intervention Efficiency Analysis

From the results of complexity analysis in Appendix B, we believe the computational complexity bottleneck of EAGLE lies in the spatio-temporal causal intervention mechanism. In this case, we analyze the intervention efficiency in the following two aspects.

**Intervention Ratio.** We perform node-wisely causal interventions as in Eq. (15). However, executing interventions for all nodes in each epoch is time-consuming. Thus, we propose randomly selecting nodes and performing interventions according to a certain ratio. Let the intervention ratio represent the ratio of the number of intervened nodes to the total number of nodes $|\mathcal{V}|$. Figure D.5 shows the changes in task performance (AUC %) and the training time as the intervention ratio increases. We observe the AUC increases, proving that the spatio-temporal causal intervention mechanism is more effective in solving the OOD generalization problem with more intervened nodes. In addition, we notice a jump in the growth rate of AUC on three datasets at the ratio of 0.6, which indicates the most suitable intervention ratio while maintaining an acceptable training time cost.

**Mixing Ratio.** The intervention set $\mathbf{s}_v$ is sampled from $\mathcal{S}_{\mathrm{ob}} \cup \mathcal{S}_{\mathrm{ge}}$. While $\mathcal{S}_{\mathrm{ob}}$ has been already prepared after we model the environments in Section 3.1, the $\mathcal{S}_{\mathrm{ge}}$ requires instantly generating, which may be a burden on the intervention efficiency. Let the maxing ratio represent the ratio of the number of observed environment samples to the number of generated environment samples. Figure D.6 shows the changes in task performance (AUC %) and the training time as the mixing ratio increases. Different from the trend in Figure D.5, AUC reached the maximum value at different ratios on the three datasets, and when the ratio is too large or small, the model performs poorly, indicating that different datasets have varying preferences for mixing ratio settings. In addition, we observe the variation in training time is not significant, verifying that although $\mathcal{S}_{\mathrm{ob}}$ needs to be generated instantly, its time cost is acceptable still, and we should pay more attention on the optimal mixing ratio.

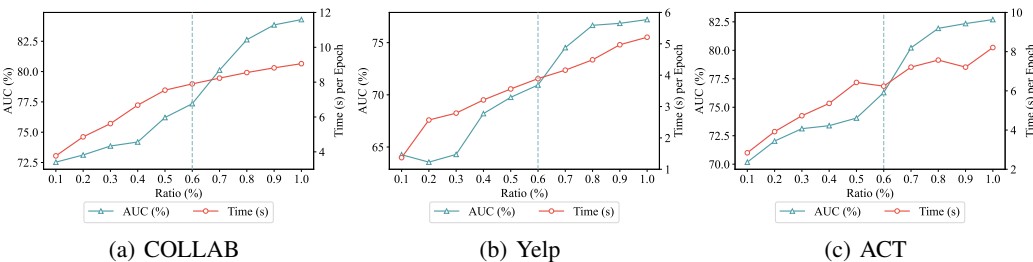

| (a) COLLAB | (b) Yelp | (c) ACT |

Figure D.5: Intervention efficiency analysis on the intervention ratio. The vertical dashed line indicates the most suitable intervention ratio while maintaining an acceptable training time cost.

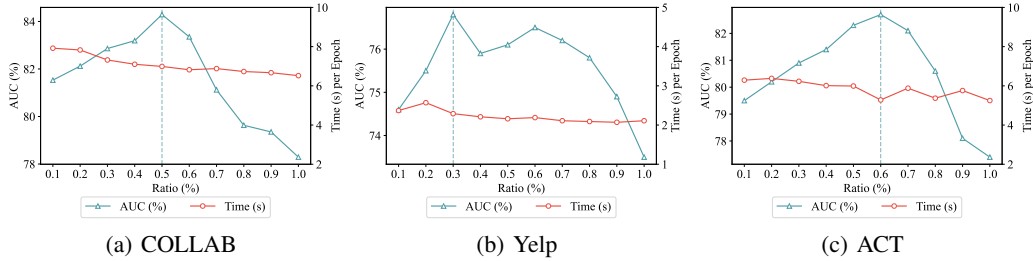

| (a) COLLAB | (b) Yelp | (c) ACT |

Figure D.6: Intervention efficiency analysis on the mixing ratio. The vertical dashed line indicates the ratio when AUC reaches the maximum value.

## D.6 Additional Analysis of Section 4.1

We visualize Table 1 and Table 2 in Section 4.1 to provide additional analysis. We have concluded in Section 4.1 that the baselines own a strong fitting ability but weak generalization ability between the distribution shifts settings. In addition to visualizing the task performance (AUC %), Figure D.7

annotates the decrease of AUC under each baseline method, where the horizontal dashed line represents the AUC decrease of our EAGLE. The smaller the decrease, the stronger the control ability under the impact of out-of-distribution shifts. We can observe that on the vast majority of datasets, our method can improve task performance in both *w/o OOD* and *w/ OOD* scenarios while minimizing AUC decrease. Our control over AUC decrease exceeds the baseline except for GAE and GCRN in the vast majority of cases. For the above two baseline methods, although they have better control ability over AUC decrease than our method, the premise is that their task performance is inherently poor. In addition, our method achieves the most excellent task performance on the ACT dataset, which can explain the unsatisfying but acceptable AUC decrease control. In summary, in addition to evaluating the advantages of our EAGLE in terms of task performance and generalization ability, which is the most topic-relative and common, our EAGLE also maintains the ability to reduce the impact of OOD on task performance.

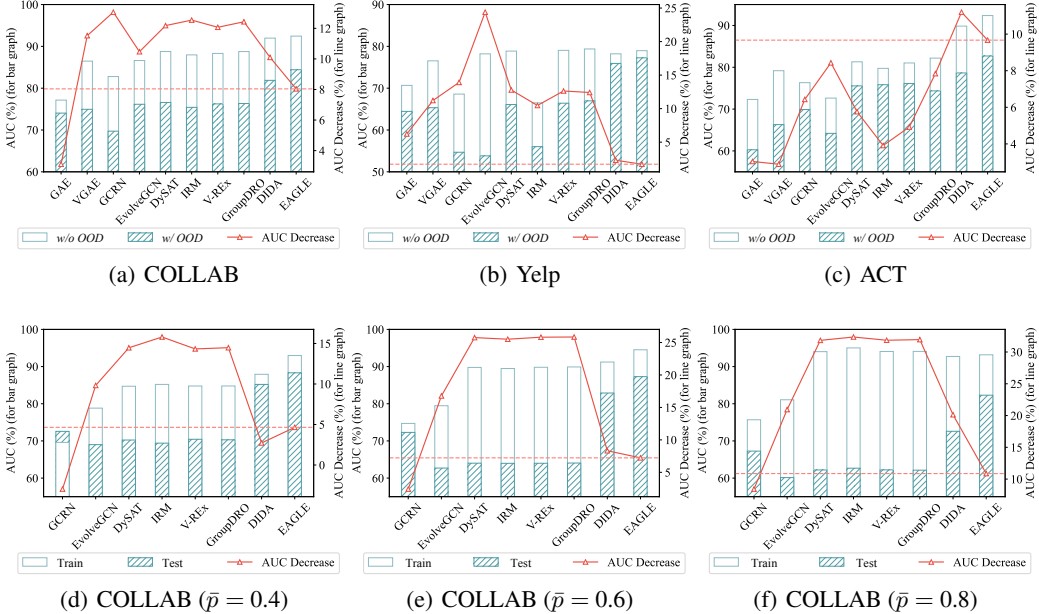

Figure D.7: Additional analysis of the performance on future link prediction.

## D.7 Additional Results of Section 4.2

Section 4.2 reports the results when $\bar{q} = 0.8$. Here we report the additional results when $\bar{q} = 0.4$ and 0.6 in Figure D.8. All detailed results are summarized in Table D.2. A similar trend can be observed as we report in Section 4.2 that as $\sigma_{\mathbf{e}}$ increases, the performance of EAGLE shows a significant increase while narrowing the gap between *w/o OOD* and *w/ OOD* scenarios. Although DIDA [94] also shows an upward trend, its growth rate is much more gradual, which indicates that DIDA [94] is difficult to perceive changes in the underlying environments caused by different $\sigma_{\mathbf{e}}$ as it is incapable of modeling the environments, thus cannot achieve satisfying generalization performance. In addition, we also notice a positive correlation between $\mathbb{I}_{\mathrm{ACC}}$ and the AUC, which verifies the improvements are attributed to the proper recognition of the invariant patterns by $\mathbb{I}(\cdot)$. In conclusion, our EAGLE can exploit more reliable invariant patterns, thus performing high-quality invariant learning and efficient causal interventions, and achieving better generalization ability.

# E Implementation Details

## E.1 Training and Evaluation

**Training Settings.** The number of training epochs for optimizing our proposed method and all baselines is set to 1000. We adopt the early stopping strategy, *i.e.*, stop training if the performance

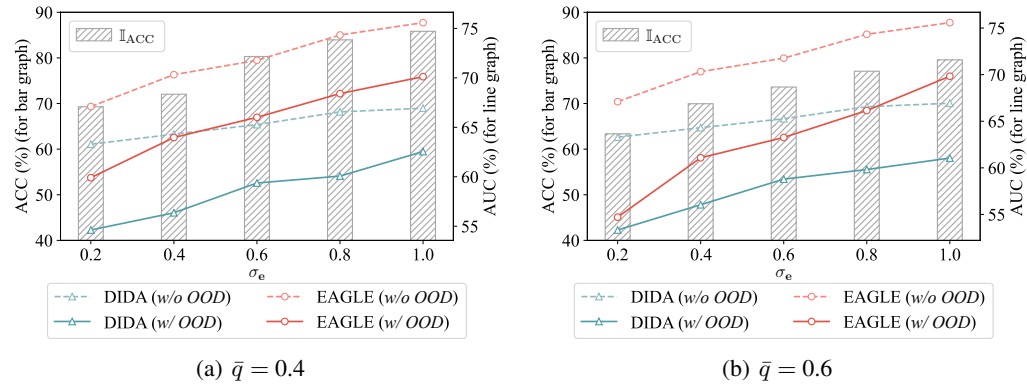

(a) $\bar{q} = 0.4$  (b) $\bar{q} = 0.6$

Figure D.8: Additional results on the effects of invariant pattern recognition.

Table D.2: AUC score (% ± standard deviation) of future link prediction task on synthetic datasets. *w/o OOD* and *w/ OOD* denote testing without and with distribution shifts. $\mathbb{I}_{\text{ACC}}$ is reported in the accuracy score). The best results are shown in **bold** and the runner-ups are underlined.

| Shift Degree | | — | $\bar{q} = 0.4$ | | $\bar{q} = 0.6$ | | $\bar{q} = 0.8$ | |
|---|---|---|---|---|---|---|---|---|
| $\sigma_{\text{e}}$ | **Model** | *w/o OOD* | *w/ OOD* | $\mathbb{I}_{\text{ACC}}$ | *w/ OOD* | $\mathbb{I}_{\text{ACC}}$ | *w/ OOD* | $\mathbb{I}_{\text{ACC}}$ |
| **0.2** | DIDA [94] | 63.29±0.35 | 54.62±0.92 | — | 53.33±1.01 | — | 52.87±1.28 | — |
| | **EAGLE** | 67.10±0.23 | 59.91±1.18 | 69.28±1.59 | 54.71±1.29 | 63.36±1.74 | 54.26±1.31 | 59.79±1.37 |
| **0.4** | DIDA [94] | 64.31±0.34 | 56.36±0.98 | — | 56.04±1.13 | — | 55.89±1.24 | — |
| | **EAGLE** | 70.32±0.27 | 63.97±0.82 | 72.04±1.34 | 61.08±1.02 | 69.95±1.30 | 58.40±1.12 | 63.88±1.36 |
| **0.6** | DIDA [94] | 65.27±0.41 | 59.37±0.87 | — | 58.79±0.97 | — | 57.35±1.19 | — |
| | **EAGLE** | 71.77±0.38 | 66.01±0.74 | 80.31±1.52 | 63.26±0.64 | 73.62±1.41 | 63.11±0.78 | 69.30±1.50 |
| **0.8** | DIDA [94] | 66.56±0.39 | 60.07±0.89 | — | 59.82±1.05 | — | 59.20±1.03 | — |
| | **EAGLE** | 74.33±0.29 | 68.41±0.72 | 83.95±1.66 | 66.15±0.69 | 77.08±1.73 | 65.82±0.81 | 72.59±1.46 |
| **1.0** | DIDA [94] | 66.93±0.18 | 62.55±0.85 | — | 61.05±0.93 | — | 60.33±1.17 | — |
| | **EAGLE** | **75.58±0.40** | **70.12±0.91** | **85.83±1.54** | **69.83±0.93** | **79.56±1.65** | **68.09±0.97** | **74.16±1.21** |

on the validation set does not improve for 50 epochs. For our EAGLE, the hyperparameter $\alpha$ is chosen from $\{10^{-3}, 10^{-2}, 10^{-1}, 10^0, 10^1\}$, and $\beta$ is chosen from $\{10^{-6}, 10^{-5}, 10^{-4}, 10^{-3}, 10^{-2}\}$. The intervention ratio and the mixing ratio are carefully tuned for each dataset. For other parameters, we adopt the Adam optimizer [40] with an appropriate learning rate and weight decay for each dataset and adopt the grid search for the best performance using the validation split. All parameters are randomly initiated, which is especially important for $\mathbf{W}_k$ in Eq. (3) that ensures the difference in each environment embedding space. The $K$ channels will still remain orthogonal during training as we conduct discrete environment disentangling iteratively. This helps the recognition of invariant/variant patterns mainly because we guarantee there is no overlap between environments.

**Evaluation.** According to respective experiment settings, we randomly split the dynamic datasets into training, validation, and testing chronological sets. We sample negative links from nodes that do not have links, and the negative links for validation and testing sets are kept the same for all baseline methods and ours. We set the number of positive links to the same as the negative links. We use the Area under the ROC Curve (AUC) [7] as the evaluation metric. As we focus on the future link prediction task, we use the inner product of a pair of learned node representations to predict the occurrence of links, *i.e.*, we implement the link predictor $g(\cdot)$ as the inner product of hidden embeddings, which is commonly applied in classic future link prediction tasks. The biased training technique is adopted following [9]. We use the cross-entropy loss as the loss function $\ell(\cdot)$. The activation function is LeakyReLU [1]. We randomly run all the experiments five times, and report the average results with standard deviations.

### E.2 Baseline Implementation Details

We provide the baseline methods implementations with respective licenses as follows.

- GAE [42]: `https://github.com/DaehanKim/vgae_pytorch` with MIT License.
- VGAE [42]: `https://github.com/DaehanKim/vgae_pytorch` with MIT License.
- GCRN [76]: `https://github.com/youngjoo-epfl/gconvRNN` with MIT License.
- EvolveGCN [62]: `https://github.com/IBM/EvolveGCN` with Apache-2.0 License.
- DySAT [75]: `https://github.com/FeiGSSS/DySAT_pytorch` with license unspecified.
- IRM [3]: `https://github.com/facebookresearch/InvariantRiskMinimization` with CC BY-NC 4.0 License.
- V-REx [44]: `https://github.com/capybaralet/REx_code_release` with license unspecified.
- GroupDRO [74]: `https://github.com/kohpangwei/group_DRO` with MIT License.
- DIDA [94]: `https://github.com/wondergo2017/DIDA` with license unspecified.

The parameters of baseline methods are set as the suggested value in their papers or carefully tuned for fairness.

### E.3  Configurations

We conduct the experiments with:

- Operating System: Ubuntu 20.04 LTS.
- CPU: Intel(R) Xeon(R) Platinum 8358 CPU@2.60GHz with 1TB DDR4 of Memory.
- GPU: NVIDIA Tesla A100 SMX4 with 40GB of Memory.
- Software: CUDA 10.1, Python 3.8.12, PyTorch [63] 1.9.1, PyTorch Geometric [20] 2.0.1.

## F  Further Discussions

### F.1  Further Analysis on SCM Model

We provide further analysis of the intrinsic cause of the out-of-distribution shifts. From the causal-based theories [64, 65, 66], we formulate the generation process of static graphs and dynamic graphs with the Structural Causal Model (SCM) [64] in Figure F.1, where the arrow between variables denotes causal dependencies. It is widely accepted in the OOD generalization works [22, 3, 72, 87, 11, 2, 60] that the correlations between labels and certain parts of the latent features are invariant across data distributions in training and testing, while the other parts of the features are variant. The invariant part is also called the causal part ($C$) and the variant part is also called the spurious part ($S$).

**Qualitative Analysis.** In the SCM model on static graphs, $C \rightarrow \mathbf{G} \leftarrow S$ demonstrates that the invariant part and variant part jointly decide the generation of the graphs, while $C \rightarrow \mathbf{Y}$ denotes the label is solely determined by the causal part. However, there exists the spurious correlation $C \dashleftarrow \dashrightarrow S$ in certain distributions that would lead to a backdoor causal path $S \dashrightarrow C \rightarrow \mathbf{Y}$ so that the variant part and the label are correlated statistically. As the variant part changes in the testing distributions caused by different environments $\mathbf{e}$, the predictive patterns built on the spurious correlations expired. For the same reason, similar spurious correlation $C^t \dashleftarrow \dashrightarrow S^t$ exists on dynamic graphs within a single graph snapshot, which opens the backdoor causal path $S^t \dashrightarrow C^t \rightarrow \mathbf{Y}^t$. Especially, as we have captured the temporal dynamics between each graph snapshot, the variant part in the previous time slice may also establish spurious correlations with the invariant part at present time, *i.e.*, $C^{t-1} \dashleftarrow \dashrightarrow S^t$, leading to $S^{t-1} \dashrightarrow C^t \rightarrow \mathbf{Y}^t$, which is a unique phenomenon in the dynamic scenarios. Hence, we propose to get rid of the spurious correlations within and between graph snapshots by investigating the latent environment variable $\mathbf{e}$, encouraging the model to rely on the spatio-temporal invariant patterns to make predictions, and thus handle the distribution shifts.

**Further Analysis of Assumption 1.** The causal inference theories [64, 65, 66] propose to get rid of the spurious correlations by blocking the backdoor path with $do$-calculus, which would remove all causal dependencies on the intervened variables. Particularly, we intervene in the variant parts on all graph snapshots, *i.e.*, $\mathrm{do}(S^t)$, and thus the spurious correlations within and between graph snapshots can be filtered out. This encourages the two conditions in Assumption 1 to be satisfied: the Invariance

Property will be satisfied if the spurious correlations $S^{t-1} \leftarrow \times \rightarrow C^t \leftarrow \times \rightarrow S^t$ are removed and the label will be solely decided by the invariant part $C^t \rightarrow \mathbf{Y}^t$, which also satisfies the Sufficient Condition. In this case, we can minimize the variance of the empirical risks under diverse potential environments, while encouraging the model to make predictions of the spatio-temporal invariant patterns.

**Further Explanations of the Toy Example.** From the above analysis, we can further explain the toy example in Figure 1(a). The prediction model has captured the spurious correlations between "coffee" and the "cold drink", which caused the false prediction of buying an Iced Americano in the winter. By applying our environment-ware EAGLE, the prediction model can perceive the environments of seasons through the neighbors around the central node, *i.e.*, perceiving the winter season by learning the observed interactions between the user and the thick clothing. Thus encouraging the model to rely on the exploited spatio-temporal invariant patterns, *i.e.*, "the user buys coffee", to make the correct prediction on the Hot Latte by considering underlying environments.

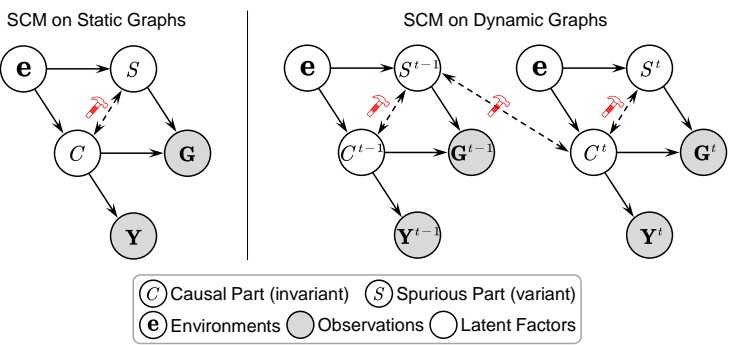

Figure F.1: The SCM model on static graphs and dynamic graphs.

## F.2 Further Understanding of Environments

Environments on dynamic graphs are latent factors, where there are no accessible ground-truth environment labels in the real world, leading to the lack of explainability. In order to improve the explainability of the environment, and patterns the EA-DGNN learns, we have provided some real-world examples to make explanations (Section 3.1). The key insight lies that, the formation of real-world dynamic graphs typically follows a complex process under the impact of latent environments, causing the relationships to be multi-attribute. To model diverse spatio-temporal environments, the ego-graphs of each node need to be disentangled and processed in different embedding spaces.

**An Easy-to-Understand Example: the Social Networks.** The relationships between the central node and its neighbor nodes, which may be classmates, colleagues, *etc.*, are formed under the influence of different surrounding environments. For example, the relationship between classmates is formed in the "school" environment, which is a compound of "classmates", "teachers", "staff", *etc.*, and the relationship between colleagues is formed in the "working" environment, *etc.* Multiple relationships are compounded into a single edge and change over time. In order to model such multiple environments, we propose multi-channel environments disentangling to represent different semantic relationships in $K$ embedding spaces. For example, the 1-st embedding represents the "classmate" relationship, and the 2-nd embedding space represents the "colleague" relationship, *etc.* Thus, EA-DGNN realizes the perception of multiple surrounding environments and encodes spatio-temporal environment information into node representations.

## F.3 Further Understanding of the Multi-Label

**Understanding.** We do not require ground-truth environment labels in our EAGLE. The environments on dynamic graphs are latent factors, where there are no accessible ground-truth environment labels that have practical meanings in the real world. This is also why conventional OOD generalization baselines on images, text, *etc.* have poor performance (Section 4.1) in graphs as they must rely on the ground-truth environment labels to generalize. In our work, the multi-label $\mathbf{y}$ in Section 3.2 is mixed up with time index $t$ and environment index $k$, indicating which environment index under which time index $\mathbf{z}$ belongs. It can be seen as our inferred label of environments.

**Example.** If there exist $K$ environments and $T$ graph snapshots, we first initialize a zero matrix of the shape $K \times T$. Then we mark the value in position $(k, t)$ to be 1 and reshape the matrix into the 1-dimension vector $\mathbf{y}$, indicating the multi-label of $\mathbf{z}$ for the $k$-th environment at time $t$.

**Role.** The multi-labels are used with $\mathbf{z}$ to infer environments by ECVAE, where the multi-label is concatenated with its corresponding $\mathbf{z}$ to realize conditional variational inference. We can then instantiate environments by generating samples from the inferred distribution with a given one-hot multi-label. This can be regarded as the data augmentation of the environment samples under the guidance of the inferred prior distribution $q_\phi(\mathbf{e} \mid \mathbf{z}, \mathbf{y})$, which helps improve the generalization ability.

## F.4 Further Understanding of Proposition 2

**Targets and Principles.** Proposition 2 provides a solution to obtain the optimal $\mathbb{I}^\star(\cdot)$ in Assumption 1 with theoretical proof in Appendix C.2. In fact, Proposition 2 solves a dynamic programming problem by optimizing the state transition equation $\mathbb{I}(i, j)$. Given $\text{Var}(\mathbf{z}_v^{\mathbf{e}\prime}) \in \mathbb{R}^K$ as the representation variance of node $v$ in each environment across times, the target is to find a partition dividing all environment patterns into invariant and variant types, so as to maximize the difference between the variance means.

**A Conceptual Example.** If $\text{Var}(\mathbf{z}_v^{\mathbf{e}\prime})$ of node $v$ is [0.1, 0.2, 0.3, 0.4, 0.9], then the optimal partition is [0.1, 0.2, 0.3, 0,4] and [0.9] (with a larger mean difference) rather than [0.1, 0.2, 0.3] and [0.4, 0.9]. An optimal partition can always be found for each node, which greatly helps the patterns discrimination and fine-grained causal interventions, improving the generalization ability, and is one of our main advantages compared with DIDA [94].

Proposition 2 intuitively illustrates a feasible implementation for the optimal $\mathbb{I}^\star(\cdot)$, providing an ideal optimizing start point. Note that, Proposition 2 itself cannot fully guarantee the global accuracy of identifying the $\mathbb{I}^\star(\cdot)$, but should optimize along with the $\mathcal{L}_{\text{risk}}$ loss (Eq.(14)), which provides theoretical ensure.

## F.5 Further Discussions Compared with DIDA

The difference between our EAGLEand DIDA [94], and EAGLE's main advantages are:

- **Modeling Environments.** EAGLE is the first to explicitly model latent environments on dynamic graphs by variational inference. DIDA [94] neglects to model complex environments, which weakens its ability to identify invariant patterns.

- **Representation Learning.** EAGLE learns node embeddings by $K$-channel environments disentangling and spatio-temporal convolutions, which helps better understand multi-attribute relations. DIDA [94] learns with single channel convolutions with an attention mechanism.

- **Invariant Learning.** EAGLE discriminates spatio-temporal invariant patterns by the theoretically supported $\mathbb{I}^\star(\cdot)$ for each node individually, leading to better removal of spurious correlations. DIDA [94] divides invariant/variant parts heuristicly with a minus operation for all nodes.

- **Causal Intervention.** EAGLE performs fine-grained causal interventions with both observed and generated environment samples, better minimizing the variance of extrapolation risks, and generalizing to unseen distributions better. DIDA [94] intervenes coarse-grainedly with only observed samples.

## F.6 Further Related Work

**Dynamic Graph Learning.** Extensive research [70, 4] address the challenges of learning on dynamic graphs, which consist of multiple graph snapshots at different times. Dynamic graph neural networks (DGNNs) are widely adopted to learn dynamic graphs by intrinsically modeling both spatial and temporal patterns, which can be divided into two main categories: spatial-first methods and temporal-first methods. The spatial-first methods [88, 28, 76] first adopt vanilla GNNs to model spatial patterns for each graph snapshot, followed by sequential-based models like RNNs [58] or LSTMs [31], to capture temporal relations. In comparison, temporal-first DGNNs [84, 73] model dynamics in advance with temporal encoding mechanisms [32], and then conduct convolutions of message-passing and aggregating on each single graph with GNNs. Dynamic graph learning has been widely utilized for prediction tasks like disease transmission prediction [38], dynamic recommender system [90],

social relation prediction [85], *etc.* However, most existing works fail to generalize under distribution shifts. DIDA [94] is the sole prior work that addresses distribution shifts on dynamic graphs with an intervention mechanism. But DIDA [94] neglects to model the complex environments on dynamic graphs, which is crucial in tackling distribution shifts. We also experimentally validate the advantage of our method compared with DIDA [94].

**Out-of-Distribution Generalization.** Most machine learning methods are built on the I.I.D. hypothesis, *i.e.*, training and testing data follow the independent and identical distribution, which can hardly be satisfied in real-world scenarios [77], as the generation and collection process of data are affected by many latent factors [71, 3]. The non-I.I.D. distribution results in a significant decline of model performance, highlighting the urgency to investigate generalized learning methods for out-of-distribution (OOD) shifts, especially for high-stake downstream applications, like autonomous driving [15], financial system [62], *etc.* OOD generalization has been extensively studied in both academia and industry covering various areas [77, 92, 30] and we mainly focus on OOD generalization on graphs [50]. Most graph-targeted works concentrate on node-level or graph-level tasks on static graphs [98, 18, 49, 86, 52, 12, 87], targeting at solving the problems of graph OOD generalization for drugs, molecules, *etc.*, which is identified as one key challenge in AI for science (AI4Science). Another main category of works elaborates systematic benchmarks for graph OOD generalization evaluation [26, 17, 36]. However, there lack of further research on dynamic graphs with more complicated shift patterns caused by spatio-temporal varying latent environments, which is our main concern.

**Invariant Learning.** Deep learning models tend to capture predictive correlations behind observed samples, while the learned patterns are not always consistent with in-the-wild extrapolation. Invariant learning aims to exploit the less variant patterns that lead to informative and discriminative representations for stable prediction [14, 46, 95]. Supporting by disentangled learning theories and causal learning theories, invariant learning tackles the OOD generalization problem from a more theoretical perspective, revealing a promising power. Disentangle-based methods [5, 55] learn representations by separating semantic factors of variations in data, making it easier to distinguish invariant factors and establish reliable correlations. Causal-based methods [22, 3, 72, 87, 11, 2, 60] utilize Structural Causal Model (SCM) [64] to filter out spurious correlations by intervention or counterfactual with *do*-calculus [65, 66] and strengthen the invariant causal patterns. However, the invariant learning method of node-level tasks on dynamic graphs is underexplored, mainly due to its complexity in analyzing both spatial and temporal invariant patterns.

**Disentangled Representation Learning.** Disentangled Representation Learning (DRL) is a paradigm that inspires a model with the capability to discriminate and disentangle latent factors intrinsic to the observable data. The fundamental objective of DRL is to separate latent factors of variation into distinct variables endowed with semantic significance. This helps to improve the generalization capacity, explainability, and robustness, *etc.* in a wide range of scenarios. Following [83], we categorize existing DRL works into traditional statistical approaches, VAE-based approaches, GAN-based approaches, hierarchical approaches, and other methods. Specifically, applying DRL to graphs results in benefits and advantages in graph tasks. DisenGCN [56], FactorGCN [89], DGCL [48], *etc.* decompose the input graph into segments for disentangled representations, which inspire our work on environment disentangling.

