# OpenReview forum: "Environment-Aware Dynamic Graph Learning for Out-of-Distribution Generalization"
_NeurIPS.cc/2023/Conference — NeurIPS 2023 poster_

### Official Review · Reviewer_aDm2 · 2023-07-02

**Soundness:** 2 fair
**Presentation:** 1 poor
**Contribution:** 2 fair
**Rating:** 4
**Confidence:** 4

**Summary:**

This paper proposes EAGLE to improve OOD generalization in DGNNs. It focuses on modeling and inferring complex environments on dynamic graphs with distribution shifts and identifying invariant patterns within inferred spatio-temporal environments. The EAGLE incorporates an environment-aware model, an environment instantiation mechanism for diversification, and an invariant pattern recognition mechanism for OOD prediction. The authors claim that EAGLE outperforms current methods on various dynamic graph datasets, marking the first approach to OOD generalization on dynamic graphs from an environment learning perspective.

**Strengths:**

1. The problem to be solved in this article is significant.
2. The authors conducted sufficient experiments to verify their claim.


**Weaknesses:**

1. The paper lacks a clear definition of 'environment', particularly in the context of multiple graphs. As the environment seems to correlate with time (T), the authors should elucidate whether the environment changes with T. Given the evident correlation between time and environmental changes - as seen in the authors' own example of transitioning from school to a company (past classmates --> current colleagues) - this should be addressed in the discussion.

2. It is not apparent whether the different environment K is determined by the dataset or is a hyperparameter. The paper should elucidate the relationship between the environment number K and the graph data T.

3. Assumption 1, which emulates a significant amount of existing work [1-4], seems to have a miswritten part (b) where the noise term should be placed outside the brackets.

4. The article uses notations that are confusing, making it difficult to read. For instance, it is unclear why the same symbol, z, is used to represent the result set obtained by the set operation in line 134. The meanings of symbols e and s should be clarified.

5. The author should provide more explanation about Proposition 1 and Proposition 2, rather than simply stating the theories and placing the proofs in the appendix. A brief explanation of what these theorems illustrate would be beneficial.

6. Several remarks and statements in the article are wrong. For example:

- The absolute statement, "Existing works fail to generalize under distribution shifts ..." overlooks recent works [4] addressing distribution shifts.
- The sentence on lines 140-141, "We regard the learned node embeddings as environment samples drawn from the ground-truth distribution of latent environment", is ambiguous and warrants clarification. How can we learn "ground-truth" distribution?

In summary, the main issue with this paper is its lack of clarity, manifested in confusing mathematical symbols, unclear theorem and formula meanings, and an ill-defined problem statement.

[1] Handling Distribution Shifts on Graphs: An Invariance Perspective, ICLR 2022

[2] Learning Invariant Graph Representations for Out-of-Distribution Generalization, NeurIPS 2022

[3] Learning Substructure Invariance for Out-of-Distribution Molecular Representations, NeurIPS 2022

[4] Dynamic Graph Neural Networks Under Spatio-Temporal Distribution Shift, NeurIPS 2022


**Questions:**

plz refer to Weaknesses.

**Limitations:**

plz refer to Weaknesses.

---

> ### Author Rebuttal · Authors · 2023-08-09
>
> We sincerely thank the reviewer for the detailed comments. Responses are as follows.
>
> **Q1.1:** The definition of “environment”.
>
> **A1.1:** Thanks for your comment. We would clarify that we have explained “environment” from the perspective of **latent factor** (lines 88-94), **real-world explanations** (lines 100-104, Figure 1(a)), and **formal definition** (line 111). In short, environments are latent factors, where the ground-truth labels are inaccessible. The impact of environments causes relationships on dynamic graphs to be multi-attribute. To model diverse spatio-temporal environments, the ego-graphs of each node are disentangled and processed in $K$ embedding spaces.
>
> ---
>
> **Q1.2:** Whether environment changes with T. Give evident correlations.
>
> **A1.2:** Thanks for your question. Environment is **time-dependent** (past classmate, current colleague). Note, time-independent is a special case, e.g., regions near the equator are always summer, shown in Figure 1(a). EAGEL doesn’t require explicit correlations between time and environment. Instead, EAGLE infers distributions of latent environments with multi-label $\mathbf{y}$, mixing with time index $t$ and environment index $k$. We will add further clarifications in the revision.
>
> ---
>
> **Q2:** How $K$ is determined? What’s the relationship between $K$ and graph?
>
> **A2:** Thanks for your question. We adopt a **warm-up mechanism** to determine $K$. We evaluate the performance of the first 10 epochs on the validation set to find the most suitable $K$, and fix it for the rest of training. We also add a parameter sensitivity experiment on COLLAB, which shows the impact of $K$ on model performance. We will add this in the revision and leave the study of how to adaptively determine $K$ for each node as future work.
>
> | $K$ | 2 | 4 | **6** | 8 | 10 |
> | --- | --- | --- | --- | --- | --- |
> | AUC (w/o OOD) | 80.33±1.29 | 81.17±1.04 | **84.41±0.87** | 83.65±0.79 | 81.94±1.03 |
>
> ---
>
> **Q3.1:** Assumption 1 emulates a significant amount of existing works [1-4].
>
> **A3.1:** Thanks for your comment. Assumption 1 is **a universally acknowledged and widely accepted assumption** in OOD works based on the Independent Causal Mechanism and Invariant Learning theory [1-4]. **However**, most related works don’t state that they can be optimized to satisfy this assumption. Proposition 3 in our work proves Assumption 1 can be satisfied by optimizing EAGLE (Appendix C.3).
>
> ---
>
> **Q3.2:** Assumption 1 seems to have a miswritten part (b).
>
> **A3.2:** Thanks for your careful and professional review. We revisit relevant literature and confirm it was a **typo**, i.e., $\epsilon$ should be placed outside the brackets. We have re-checked that the typo will not affect Assumption 1 or the related propositions and proofs. We will correct this in the revision.
>
> ---
>
> **Q4.1:** Notations are confusing and difficult to read. Why $\mathbf{z}$ is used to represent results of set operation?
>
> **A4.1:** Thanks for your questions. We use $\mathbf{z}$ to denote “embeddings”. $\mathbf{z}$ with different super/subscripts introduces extra meaning. Explanations for all $\mathbf{z}$ are listed in Appendix A. To avoid confusion, we will update $\mathbf{z}$ in line 134 to $\mathbf{Z}$ in the revision. We will further clarify the notations.
>
> ---
>
> **Q4.2:** The meanings of $\mathbf{e}$ and $s$.
>
> **A4.2:** Thank you for your question.
>
> - $\mathbf{e}$ means multiple latent environments (lines 88-89). We explicitly define $\mathbf{e}_v$ = {$\mathbf{e}_k$}$_1^K$ in line 111, which means the multiple latent environment of node $v$ is a compound of $K$ surrounding environments.
> - $s$ is the element in $\mathcal{S} _{ob} \cup \mathcal{S} _{ge}$ (Eq. (15)), i.e., the instances sampled from the observed and generated environment samples.
>
> Detailed explanations for $\mathbf{e}$ and $s$ are listed in Appendix A.
>
> ---
>
> **Q5:** More explanation about Proposition 1 and Proposition 2.
>
> **A5:** Thank you for your suggestions.
>
> - **[Proposition 1]** introduces loss function $\mathcal{L}_{\mathrm{ECVAE}}$ (Eq. (8)) to train ECVAE. ECVAE is realized by fully connected layers with the optimization goal in Eq. (C.6). To make it tractable, we apply MCMC sampling and reparameterization tricks, and reach the final optimization goal for ECVAE as Eq. (8).
> - **[Proposition 2]** solves a dynamic programming problem to obtain $\mathbb{I}^\star(\cdot)$ with proofs in Appendix C.2. The target is to find a partition dividing all patterns into (in)variant set with the max difference between the variance means. Suppose $\mathrm{Var}(\mathbf{z}_v^{\mathbf{e}\prime})$ is [0.1, 0.2, 0.3, 0.4, 0.9], then the optimal partition is [0.1, 0.2, 0.3, 0,4] and [0.9] (with a larger mean difference) rather than [0.1, 0.2, 0.3] and [0.4, 0.9]. An optimal partition can always be found for each node, which contributes to the generalization.
>
> We will provide further explanations in the revision.
>
> ---
>
> **Q6:** The absolute statement, “Existing works fail to generalize under distribution shifts ...” overlooks recent work DIDA addressing distribution shifts.
>
> **A6:** Thanks for your comment. We clarify that our original sentence is “**most** existing works fail to generalize under distribution shifts. **DIDA [84] is the sole prior work** that tackles ......” (lines 303-305). We also express the exactly same meaning in lines 427-431 (Appendix F.2). We respectfully think our sentence clearly states the related works and is not an absolute statement.
>
> ---
>
> **Q7:** Sentence on lines 140-141 is ambiguous. How can we learn “ground-truth” distribution?
>
> **A7:** Thanks for your question. We clarify that this sentence does not imply EAGLE could learn the ground-truth distribution. By “**regard**” the node embeddings obtained by EA-DGNN as environment samples drawn from the ground-truth distribution, we can **infer** the ground-truth distribution with the observed samples by ECVAE. We will further clarify our paper to avoid potential misunderstandings.

---

> > ### Comment · Reviewer_aDm2 · 2023-08-16
> >
> > Thank you author for the rebuttal. I found some concerns after reading this article and DIDA [1] carefully.
> >
> > 1. There are a lot of over-declarations in the article, such as "this is the first trial to explore the impact of environments on dynamic graphs under OOD shifts. However, the OOD problem on dynamic graphs has been studied and defined by other works, such as DIDA.
> > And DIDA has also explored the environment or variant patterns. Therefore, the contribution of the article is insufficient.
> > 2. In addition, a lot of content is imitated from the work of DIDA, such as the problem description in Section 2, and Assumption 1.
> > So I think this part of the contribution is also insufficient.
> > 3. I am also very surprised why there is no detailed comparison with DIDA in this article. For example, what are the problems in the work of DIDA, and what are the advantages of this work compared with DIDA?
> >
> > In summary I do not decide to revise my score.
> >
> > [1] Dynamic Graph Neural Networks Under Spatio-Temporal Distribution Shift, NeurIPS 2022

---

> > > ### Author Response · Authors · 2023-08-17
> > >
> > > We thank the reviewer for the discussion. After carefully reading the reviewer's comments, we think there exist some misunderstandings and would like to clarify the concerns point by point.
> > >
> > > **Q1.1:** There are a lot of over-declarations in the article, such as “this is the first trial to explore the impact of environments on dynamic graphs under OOD shifts”. However, the OOD problem on dynamic graphs has been studied and defined by other works, such as DIDA.
> > >
> > > **A1.1:** Thank you for your concerns and comment.
> > >
> > > - We would like to claim that, we **do** propose to **explore the impact of environments** on dynamic graphs under OOD shifts **for the first time**, by applying the novel Environment “Modeling-Inferring-Discriminating-Generalizing” paradigm of our EAGLE.
> > > - Our above contribution **does not imply** we are the first to study the problem of OOD on dynamic graphs, as **we reiterated that DIDA [1] is the sole prior work that tackles distribution shifts on dynamic graph** (e.g., lines 304-305), but **we are the first to utilize “environment” to solve the problem**. Specifically, we tackle this problem from a **novel** perspective (the latent environments) and gain **better** performance due to our **advantages** compared to DIDA [1] (please see A3 for details).
> > >
> > > In summary, these declarations emphasize the contribution of investigation into “environments” (first time), rather than works on the dynamic OOD problem.
> > >
> > > ---
> > >
> > > **Q1.2:** And DIDA has also explored the environment or variant patterns. Therefore, the contribution of the article is insufficient.
> > >
> > > **A1.2:** Thank you for your comment.
> > >
> > > - In terms of “environment”, we have rechecked DIDA [1] carefully and found there are **no investigations on “environment” or other similar terms**.
> > > - **The variant patterns exploited by DIDA [1] are different from the “environments” in our work**. Specifically, DIDA [1] defines variant/invariant patterns as subsets of ego-graphs across time stamps whose predictively to labels are (not) stable across time periods and graph communities **(all in only one embedding space)**, while our explanations for “environments” are $K$ **latent embedding spaces**, which are formed under the impacts of complex spatio-temporal node interactions, causing the relationships to be multi-attribute.

---

> > > ### Author Response · Authors · 2023-08-17
> > >
> > > (continue)
> > >
> > >
> > > **Q2:** In addition, a lot of content is imitated from the work of DIDA, such as the problem description in Section 2, and Assumption 1. So I think this part of the contribution is also insufficient.
> > >
> > > **A2:** Thank you for your comment.
> > >
> > > - **[Section 2]** First, we would like to clarify that **we never mentioned the “problem” defined in Section 2 is new or it should be considered as our contribution**. We have cited DIDA [1] accurately and comprehensively in the main paper and Appendix. However, **we tackle the problem, i.e., OOD generalization on dynamic graphs, from a totally different perspective with a novel method**, enjoying superior advantages and achieving better results.
> > > - **[Assumption 1]** We have replied to this concern in the rebuttal (please see Q3.1 and A3.1 for details). We would like to explain it again. Assumption 1 is **a universally acknowledged and widely accepted assumption** in almost all works based on the Independent Causal Mechanism (ICM) and the Invariant Learning theory, not only in graph OOD related works [1-4], but also in works of other fields [5-7]. So we do not consider this as imitations or emulations of DIDA [1] or our main contributions. Besides, most related works **do not** provide the support that their models can be optimized to satisfy this assumption. **We propose Proposition 3 supporting that optimizing EAGLE can help satisfy Assumption 1, which is our contribution**. We further provide their proofs in Appendix C.3.

---

> > > ### Author Response · Authors · 2023-08-17
> > >
> > > (continue)
> > >
> > > **Q3:** I am also very surprised why there is no detailed comparison with DIDA in this article. For example, what are the problems in the work of DIDA, and what are the advantages of this work compared with DIDA?
> > >
> > > **A3:** Thank you for your comment. We would like to point out that **we have provided detailed comparisons with DIDA [1] in the rebuttal and we will add the comparison in the revised version**. We also explain the comparisons again as follows.
> > >
> > > - **[Problems]** We acknowledge that we solve the same problem and follow the same task settings proposed by DIDA [1], which is not considered our contribution. We have made citations accurately and comprehensively in the contents. However, we tackle the problem from a new perspective with a novel framework, achieving better results.
> > > - **[Advantages]** Due to the page limit, we briefly discuss the comparison with DIDA in the main paper (lines 270-271) and experimentally validate the advantage of our
> > > method compared with DIDA [1]. The detailed difference and EAGLE’s advantages are:
> > >     - **[Modeling environments]** EAGLE is the **first** to explicitly model latent environments on dynamic graphs by variational inference. DIDA [1] neglects to model complex environments, which weakens its ability in identifying invariant patterns.
> > >     - **[Representation learning]** EAGLE learns node embeddings by $K$-channel environments disentangling and spatio-temporal convolutions, which helps better understand multi-attribute relations. DIDA [1] learns with single channel convolutions with attentions.
> > >     - **[Invariant learning]** EAGLE discriminate spatio-temporal invariant patterns by **the theoretically supported $\mathbb{I}^\star(\cdot)$** for each node individually, leading to better removal of spurious correlations. DIDA [1] divides invariant/variant parts simply and heuristically with a minus operation for all nodes.
> > >     - **[Causal intervention]** EAGLE perform fine-grained causal interventions with **both** observed and generated environment samples, better minimizing the variance of extrapolation risks, generalizing to unseen distributions better. DIDA [1] intervenes coarse-grainedly with only observed samples.
> > >
> > > ---
> > >
> > > We will make sure to provide the necessary clarifications and avoid these misunderstandings in the revision. Should you have any further questions or concerns, we are glad to continue the discussion and provide further clarifications.
> > >
> > > [1] Dynamic graph neural networks under spatio-temporal distribution shift. NeurIPS, 2022.
> > >
> > > [2] Learning invariant graph representations for out-of-distribution generalization. NeurIPS, 2022.
> > >
> > > [3] Learning substructure invariance for out-of-distribution molecular representations. NeurIPS, 2022.
> > >
> > > [4] Handling distribution shifts on graphs: An invariance perspective. ICLR, 2022.
> > >
> > > [5] Out-of-distribution generalization with causal invariant transformations. IEEE CVPR, 2022.
> > >
> > > [6] Causal inference by using invariant prediction: identification and confidence interval. Journal of the Royal Statistical Society Series B: Statistical Methodology, 2016.
> > >
> > > [7] Invariant risk minimization. arXiv, 2019.

---

> > > > ### Comment · Reviewer_aDm2 · 2023-08-18
> > > >
> > > > Thanks, I have read the author's rebuttal and I decided to keep my score unchanged.

---

> > > > > ### Author Response · Authors · 2023-08-20
> > > > >
> > > > > Thanks for the time and effort in reviewing our work and we would like to express our sincere gratitude to you for providing constructive suggestions. Hope our additional experiments and clarifications have addressed your concerns. We are happy to discuss with you further if you have other questions.

---

### Official Review · Reviewer_YPMZ · 2023-07-06

**Soundness:** 3 good
**Presentation:** 3 good
**Contribution:** 3 good
**Rating:** 7
**Confidence:** 4

**Summary:**

This paper proposes a novel framework called EAGLE to address the challenge of out-of-distribution generalization in dynamic graphs. The authors investigate the impact of latent environments on dynamic graphs and develop methods to model and infer these environments. They propose techniques to recognize spatio-temporal invariant patterns and perform fine-grained causal interventions. The rationality and correctness of EAGLE are guaranteed by rigorous reasoning and mathematical proof. Experimental results demonstrate the superiority of their method in handling distribution shifts in dynamic graph datasets.

**Strengths:**

(1) First attempt to explicitly model the environment on dynamic graphs. Through the innovative Environment “Modeling-Inferring-Discriminating-Generalizing” paradigm, the influence of open environment on OOD generalization on dynamic graphs is explored, and the generalization ability of the model in this scenario is improved.
(2) Well-organized derivation and mathematical proofs. Based on causality theories, the proposed method is proven to satisfy the invariance assumptions and propositions, directing the optimization that achieves OOD generalization for in-the-wild extrapolations by fine-grained causal interventions.
(3) Reasonable experiment settings and sufficient results. Appropriate baseline methods are compared under multiple OOD environment settings. The results of the main experiment and several auxiliary experiments demonstrate the effectiveness and superiority of the proposed method.
(4) This method can be easily extended to other sequential models to perform temporal convolutions, or compounded with any attention/reweighting mechanisms in environment modeling. This paper is well-organized and easy to follow. The background problems and method design are clearly explained.


**Weaknesses:**

(1) Variable symbols are kind of complicated, and their expressions are not very clear, which brings difficulties in understanding theoretical proofs.
(2) Computational complexity and sufficient analysis are not discussed in the main contents.
(3) The distribution shifts in the experiment datasets are manually designed. I wonder if there exist naturally formed dynamic graph OOD datasets that measure on their OOD degree, and conduct experiments based on those datasets.


**Questions:**

(1) Are there other OOD types besides the links and node features? How to demonstrate that the proposed model improves generalization performance on the other OOD types?
(2) It is better to explain the details of the baselines and datasets, like the differences with the proposed model, the statistical information of the datasets, etc. in the main contents.


**Limitations:**

Y

---

> ### Author Rebuttal · Authors · 2023-08-09
>
> We sincerely thank the reviewer for the detailed comments and insightful questions. We make responses to the reviewer’s comments as follows.
>
> **Q1:** Variable symbols are kind of complicated, and their expressions are not very clear, which brings difficulties in understanding theoretical proofs.
>
> **A1:** Thanks for your kind suggestion. In order to enhance readability and reduce the difficulty of understanding symbols, we will improve complex symbols and notations by increasing the descriptions and omitting unnecessary superscripts and subscripts. Especially in the theoretical proofs, we will add easy-to-follow understandings and examples in the revision to improve the reader experience.
>
> ---
>
> **Q2:** Computational complexity and sufficient analysis are not discussed in the main contents.
>
> **A2:** Thanks for your suggestion.
>
> - **[Computational complexity]** We include the computational complexity and its analysis in Appendix B. The computational complexity of EAGLE is $\mathcal{O} \left ( |\mathcal{E}| \sum_{l=0}^{L} d^{(l)}+ \mathcal{V} \left ( \sum_{l=1}^{L} d^{(l-1)}d^{(l)} + (d^{(L)})^2 \right ) \right )$ (Eq. (B.1)), indicating EAGLE has **a linear computation complexity** with respect to the number of nodes and edges, which is **on par with** existing dynamic GNNs and OOD methods on dynamic graphs.
> - **[Space complexity]** Denote $|\mathcal{V}|$ and $|\mathcal{E}|$ as the number of nodes and edges, respectively, $K$ as the number of environments, $T$ as the number of time slices, $L$ as the number of layers in EA-DGNN, $L^\prime$ as the number of layers in ECVAE, $d_0$ as the dimension of input node features, $d=K d^\prime$ as the hidden dimension of EAConv layers in EA-DGNN, $d^\prime$ as the hidden dimension of the encoder and decoder networks layer of ECVAE, $\sum_{v \in \mathcal{V}} \mathrm{Var}(\mathbf{z}_v^{\mathbf{e}\prime})$ as the variance of $K$ environment-aware representations.
>     - the dynamic graph: $\mathcal{O}(K T(|\mathcal{V}|+|\mathcal{E}|))$
>     - the node input features: $\mathcal{O}(|\mathcal{V}|  K  T  d_0)$
>     - the EAConv layer: $\mathcal{O}(L  d^2)$
>     - ECVAE: $\mathcal{O}(L^\prime  d^\prime)$
>     - storing generated environment samples: $\mathcal{O}(|\mathcal{V}| K  T  d^\prime)$
>     - storing the states for function $\mathbb{I}(\cdot, \cdot)$: $\mathcal{O}(K  \sum_{v \in \mathcal{V}} \mathrm{Var}(\mathbf{z}_v^{\mathbf{e}\prime}))$.
>     - The **overall** space complexity of EAGLE can be roughly calculated as $\mathcal{O}(K T(|\mathcal{V}|+|\mathcal{E}|)) + \mathcal{O}(|\mathcal{V}|  K  T  d) + \mathcal{O}(L  d^2) + \mathcal{O}(L d) + \mathcal{O}(K  \sum_{v \in \mathcal{V}} \mathrm{Var}(\mathbf{z}_v^{\mathbf{e}\prime}))$ (note that, we omit differences in superscript and subscript of $L$’s and $d$’s for brevity)
>
> As $K$, $T$, $d$ and $L$ are small numbers, EAGLE has a linear space complexity with respect to $|\mathcal{V}|$ and $|\mathcal{E}|$. Our experiments show that the empirical memory cost of EAGLE **is on par with the related works.** We will add these analyses in the main contents in the revised version.
>
> ---
>
> **Q3:** If there exist naturally formed dynamic graph OOD datasets that measure on their OOD degree, and conduct experiments based on those datasets.
>
> **A3:** Thanks for your question. To the best of our knowledge, there is no existing dynamic graph OOD datasets that measure their OOD degree.
>
> - **[Why no naturally formed OOD datasets]** Out-of-distribution shifts naturally exist in dynamic graphs as the formation of real-world dynamic graphs typically follows a complex process under the impact of underlying environments. However, though these shifts vary in degree, they are **difficult to quantify**. We leave the problem of measuring the OOD degree between training and testing distributions as future work.
> - **[Our datasets are more challenging]** In our experiments, we manipulate the datasets following DIDA [1]. Actually, the manually designed distribution shifts in our work are **more practical and challenging** in real-world scenarios as the model **cannot get access to any information** about the filtered links until the testing stages. Our model still has satisfactory performance under such harsh OOD distribution shifts than all baselines, which further demonstrates EAGLE's excellent out-of-distribution generalization ability.
>
> ---
>
> **Q4:** Are there other OOD types besides the links and node features? How to demonstrate that the proposed model improves generalization performance on the other OOD types?
>
> **A4:** Thanks for your question. Though there exist other OOD types on the graph level, such as the graph size, scaffold, base motifs, etc., none of them has a time dimension (not dynamic) and has no real-world scenarios in terms of the dynamic settings, leaving them **only suitable for static graphs**. As our work focuses on the OOD generalization problem on dynamic graphs for **node-level tasks**, the most related OOD types on the node level are the **links and node features**. So we conduct extensive experiments to demonstrate the generalization ability of our model in node-level OOD shift settings.
>
> ---
>
> **Q5:** Explain the details of the baselines and datasets in the main contents.
>
> **A5:** Thank you for your suggestions.
>
> - **[Baselines]** We provide a short introduction in the main contents (Section 4, lines 220 to 227) and provide the full details of the baseline and the analysis of the differences with our proposed model in Appendix D.2, and the implementation details of the baseline are introduced in Appendix E.2.
> - **[Datasets]** We give a simplified explanation and the OOD settings in the main contents (Section 4, lines 216 to 219), and implement the full details of the baselines in Appendix D.1, including statistical information, the OOD manipulations, their visualizations, etc.
>
> We will add more details of baselines and datasets in the main contents in the revision, to enhance a better understanding of experiments.

---

> > ### Comment · Reviewer_YPMZ · 2023-08-18
> >
> > Thanks the authors for their careful rebuttals and efforts, which have resolved all my questions and concerns, and proposed corresponding improvement in the revision. In summary, this paper proposes a novel framework to address the challenge of out-of-distribution generalization in dynamic graphs. The authors investigate the impact of latent environments on dynamic graphs for the first time and extensive experiments demonstrate their superiors compared to baselines. It is worth noting that, OOD generalization on dynamic graphs is under-explored, and this work emphasizes the importance of the latent environments, which I think is insightful for the upcoming works. I suggest this paper to be accepted.

---

> > > ### Author Response · Authors · 2023-08-20
> > >
> > > Dear Reviewer YPMZ,
> > >
> > > We would like to express our sincere gratitude to you for endorsing our work and providing constructive suggestions. OOD generalization on dynamic graphs is an interesting but under-explored task, and our work investigates the impact of latent environments on dynamic graphs for the first time, which brings a new perspective to the upcoming work in this field. Thanks again for the time and effort in reviewing our work!

---

### Official Review · Reviewer_iWRC · 2023-07-06

**Soundness:** 3 good
**Presentation:** 3 good
**Contribution:** 3 good
**Rating:** 5
**Confidence:** 4

**Summary:**

In this paper, the authors propose a novel framework EAGLE, Environment-Aware dynamic Graph LEarning, which tackles the OOD generalization problem by modeling complex dynamic environments and exploiting spatial-temporal invariant patterns. Following the Environment “Modeling-Inferring-Discriminating-Generalizing” paradigm, EAGLE consists of three modules. Firstly, an Environment-Aware Deep Graph Neural Network (EA-DGNN) is designed to model environments by learning disentangled representations under multi-channel environments. EAGLE then diversifies the observed environments' samples with the environment instantiation mechanism, which infers environment distributions by applying the multi-label variational inference. Finally, EAGLE discriminates spatio-temporal invariant patterns for out-of-distribution prediction by the invariant pattern recognition mechanism and performs fine-grained causal interventions node-wisely with a mixture of observed and generated environment samples. Extensive experiments on real-world datasets demonstrate that EAGLE outperforms baseline methods on dynamic graph OOD tasks.

**Strengths:**

1. The OOD generalization problem on dynamic graphs is important, especially when considering changes in latent environments. EAGLE is the first model to tackle this problem by modeling the impact of environments on dynamic graphs under distribution shifts.
2. The proposed Environment “Modeling-Inferring-Discriminating-Generalizing” paradigm is technically sound and easy to understand, and using the proposed Environment Instantiation Mechanism to generate instances for inferring environment is novel.
3. The experiments are extensive (real-world datasets under distribution shifts on link attributes and node features, ablation studies on proposed mechanisms), and the results are considerably better than existing methods.

**Weaknesses:**

1. The requirement of the environment information is not specified. Especially, whether EAGLE requires the environment label to be known or not. The reviewer thinks one of the main contributions of EAGLE should be inferring environments from samples, but there are not any detailed explanations, and it seems that the number of environments is pre-determined in the experimental setting. The reviewer wonders if the authors could discuss the requirement of environment labels in detail and how to select the proper number of environments if the environment label is unknown.
2. Proposition 2 is difficult to understand and lacks details about how to obtain the optimal $\mathbb{I}^*(\cdot)$. It would be great if the authors could provide further explanation and a concrete example to help readers better understand the proposition.
3. The related work part should also add literature related to disentangled representation learning and further discuss the difference between EAGLE and DIDA [61] in detail.

Minor issues:
typo: line 145, multi-lables -> multi-labels

**Questions:**

1. Please see the first two questions in Weaknesses.
2. Compared to DIDA, what is the advantage of EAGLE for modeling the complex dynamic environment? It would be great if the authors could provide some examples for explanation.
3. When the training data is homogeneous (i.e., only contains few environments), how does EAGLE generalize to unseen test environments with OOD shifts?
4. How to select an appropriate number of generated instances for Invariant Pattern Recognition to exploit spatio-temporal invariant patterns?
5. The authors discussed EAGLE's computational complexity in the appendix. What is EAGLE's space complexity?

**Limitations:**

The authors did not address the limitations of their work.  The reviewer thinks that the authors should make the description of the problem setting, the definitions of spatio-temporal invariant patterns, and variant patterns more clear.

---

> ### Author Rebuttal · Authors · 2023-08-08
>
> We sincerely thank the reviewer for the detailed comments. Responses are as follows.
>
> **Q1.1:** Whether EAGLE requires the environment label?
>
> **A1.1:** Thanks for your question. EAGLE doesn’t require the ground-truth environment labels, which are also **inaccessible** in the real dynamic graphs. We propose a multi-label $\mathbf{y}$, which is mixed up with time and environment index. It can be seen as our inferred environment labels.
>
> ---
>
> **Q1.2:** How to select the proper number of environments?
>
> **A1.2:** Thanks for your question. We adopt a **warm-up mechanism** to determine $K$. We evaluate the performance of the first 10 epochs on the validation set to find the most suitable $K$, and fix it for the rest of training. Following your suggestion, we add the parameter sensitivity experiment for $K$ on COLLAB, which shows the impact of $K$ on model performance. We will add this in the revision.
>
> | $K$ | 2 | 4 | **6** | 8 | 10 |
> | --- | --- | --- | --- | --- | --- |
> | AUC (w/o OOD) | 80.33±1.29 | 81.17±1.04 | **84.41±0.87** | 83.65±0.79 | 81.94±1.03 |
>
> ---
>
> **Q2:** Further explanation and example to understand Proposition 2.
>
> **A2:** Thank you for your suggestion.
>
> - **[Explanation]** Proposition 2 solves a **dynamic programming problem** to obtain the optimal $\mathbb{I}^\star(\cdot)$ with proofs in Appendix C.2. The **target** is to find a partition dividing all patterns into variant/invariant set with the **maximum** difference between the variance means.
> - **[Example]** Suppose $\mathrm{Var}(\mathbf{z}_v^{\mathbf{e}\prime})$ is [0.1, 0.2, 0.3, 0.4, 0.9], then the optimal partition is [0.1, 0.2, 0.3, 0,4] and [0.9] (with a larger mean difference) rather than [0.1, 0.2, 0.3] and [0.4, 0.9]. An optimal partition **can always be found** for each node, which greatly improves the generalization, and is **one of our main advantages** compared with DIDA [1].
>
> We will provide further explanations in the revision.
>
> ---
>
> **Q3:** More related work to disentangled representation learning.
>
> **A3:** Thank you for your valuable suggestions. Disentangled Representation Learning (DRL) is closely related to our work. Most existing graph OOD works fail to learn separate semantics of latent and complex environments, and DRL inspires us to perform environment disentangling, which greatly helps to improve the generalization capability. We will update this in the revision to enhance the understanding of environments.
>
> ---
>
> **Q4:** Further discuss the difference and between EAGLE and DIDA [1]. What are EAGLE’s advantages?
>
> **A4:** Thanks for your constructive suggestion. The difference and advantages are:
>
> - **[Modeling environments]** EAGLE is the **first** to explicitly model latent environments on dynamic graphs by variational inference. DIDA [1] neglects to model environments, which weakens its ability in identifying invariant patterns.
> - **[Representation learning]** EAGLE learns node embeddings by $K$-channel environments disentangling and spatio-temporal convolutions, which helps better understand multi-attribute relations. DIDA [1] learns with single channel convolutions with attention.
> - **[Invariant learning]** EAGLE discriminate spatio-temporal invariant patterns by **the theoretically supported $\mathbb{I}^\star(\cdot)$** for each node individually, leading to better removal of spurious correlations. DIDA [1] divides invariant/variant parts simply and heuristically with a minus operation for all nodes.
> - **[Causal intervention]** EAGLE perform fine-grained causal interventions with **both** observed and generated environment samples, better minimizing the variance of extrapolation risks, generalizing to unseen distributions better. DIDA [1] intervenes coarse-grainedly with only observed samples.
>
> We will add the comparison in the revision.
>
> ---
>
> **Q5:** How does EAGLE generalize to unseen test environments with OOD shifts when training data is homogeneous?
>
> **A5:** Thank you for your question. EAGLE is agnostic about the intrinsic properties of data (density, heterophily, etc.).  Further, following OOD literature, we assume there are $K$ latent environments, where $K$ can be determined by **the warm-up mechanism**. We leave the study of how to dynamically and adaptively determine $K$ for different datasets as our future works.
>
> ---
>
> **Q6:** How to select an appropriate number of generated instances for Invariant Pattern Recognition?
>
> **A6:** Thank you for your question. In Appendix D.5, we provide analysis and appropriate mixing ratio settings. As the number of the observed instances is fixed ($n_{ob}=|\mathcal{V}| \times K \times T$), the number of generated instances can be calculated by multiplying the preferred ratio with $n_{ob}$.
>
> ---
>
> **Q7:** What is EAGLE's space complexity?
>
> **A7:** Thank you for your question. The overall space complexity of EAGLE is $\mathcal{O}(KT(|\mathcal{V}|+|\mathcal{E}|)) + \mathcal{O}(|\mathcal{V}|KTd) + \mathcal{O}(Ld^2) + \mathcal{O}(Ld)+ \mathcal{O}(K\sum_{v\in\mathcal{V}} \mathrm{Var}(\mathbf{z}_v^{\mathbf{e}\prime}))$, where the meaning of notations can be found in Appendix A and B. As $K$, $T$, $d$ and $L$ are small numbers, EAGLE has a linear space complexity with respect to $|\mathcal{V}|$ and $|\mathcal{E}|$. Our experiments also show that the empirical memory cost of EAGLE is on par with the related works. We will add the detailed space complexity in the revision.
>
> ---
>
> **Q8.1:** No limitations statements.
>
> **A8.1:** Thanks for your suggestions. We would like to clarify that we have briefly discussed the limitations of our work being restricted only to node-level tasks **in Section 6, lines 322-323.**  We will extend limitation discussions in the revision.
>
> ---
>
> **Q8.2:** Make the descriptions more clear.
>
> **A8.2:** Thank you for your suggestion. We will make further improvements with clearer notations and easy-to-understand explanations in the revision.
>
> ---
>
> [1] Dynamic graph neural networks under spatio-temporal distribution shift. NeurIPS, 2022.

---

> > ### Comment · Reviewer_iWRC · 2023-08-19
> > **Re: Rebuttal by Authors**
> >
> > Thank you for the detailed rebuttal. I appreciate the authors providing the experimental results for selecting the appropriate $K$. However, I feel like the warm-up mechanism cannot fully convince me, as we usually do not know which kind of features (i.e., invariant or spurious) are learned by the model in the first 10 epochs, and the experimental results demonstrate that different $K$ affects the AUC score. Therefore, I would like to keep my score unchanged.

---

> > > ### Author Response · Authors · 2023-08-20
> > >
> > > We thank the reviewer for the feedback. We would like to further clarify your concerns as follows.
> > >
> > > **Q:** I feel like the warm-up mechanism cannot fully convince me, as we usually do not know which kind of features (i.e., invariant or spurious) are learned by the model in the first 10 epochs, and the experimental results demonstrate that different $K$ affects the AUC score.
> > >
> > > **A:** Thanks for your question. $K$ reflects the number of underlying environments, which is closely related to the datasets.  Thus, we indeed need to decide the most appropriate $K$ for each dataset.  As a general practice, we treat $K$ as a hyper-parameter and tune it on the validation set for the first 10 epochs, as we explained in the last response. Following your suggestion, we further demonstrate that 10 epochs are sufficient to decide the most appropraite $K$ empirically. Specifically, we show the AUC on COLLAB dataset with different $K$ as follows:
> > >
> > > | $K$ | 2 | 4 | 6 | 8 | 10 |
> > > | --- | --- | --- | --- | --- | --- |
> > > | Epoch 2 | 61.07 | 62.79 | 63.91 | 63.11 | 60.22 |
> > > | Epoch 4 | 63.44 | 64.32 | 65.23 | 64.60 | 61.34 |
> > > | Epoch 6 | 64.21 | 67.51 | 67.79 | 67.35 | 62.98 |
> > > | Epoch 8 | 66.58 | 68.23 | 69.95 | 68.83 | 64.05 |
> > > | Epoch 10 | 68.29 | 69.96 | 71.26 | 70.24 | 66.12 |
> > > | Final Epoch (converged) | 80.33 | 81.17 | 84.41 | 83.65 | 81.94 |
> > >
> > > The above results show that, though not converged, the model performance for the first 10 epochs is able to determine the optimal $K$ value, which is **consistent with the final result, validating the effectivness of our warm-up mechanism.** We will incorporate the results in the revision.
> > >
> > > Besides, we would like to clarify that the warm-up mechanism is a training “trick” we adopted in practice to improve the training efficiency, and we do not consider it as a main contribution. **There’s no big difference to fine-tune** $K$ **with complete epochs without applying the warm-up mechanism.** We will leave exploring more advanced methods to decide $K$ as future works.
> > >
> > > Should you have any further questions or concerns, we are glad to provide further responses.

---

> > > ### Author Response · Authors · 2023-08-21
> > >
> > > Dear Reviewer iWRC,
> > >
> > > As we draw closer to the rebuttal deadline, I would like to inquire if you have any additional questions or concerns about our work. We greatly value your feedback. Thank you!
> > >
> > >
> > > Best,
> > >
> > > Authors from submission 1276

---

> > > > ### Comment · Reviewer_iWRC · 2023-08-21
> > > > **Re: Official Comment by Authors**
> > > >
> > > > Thank the authors for their response and efforts. I agree that the warm-up mechanism for selecting appropriate $K$ is not the main contribution of the paper, but I feel it still has a significant impact on the model performance, as it reflects the number of environments. Therefore, without further improvement on the mechanism for selecting appropriate $K$, I would like to keep my score as is.

---

> > > > > ### Author Response · Authors · 2023-08-21
> > > > >
> > > > > Thanks for your question. As we regard $K$ a hyperparameter, its impact on the model performance demonstrates our understanding and investigation of the latent environments on dynamic graphs is rational. Though choosing the appropriate $K$ by fine-tuning is a common way, we will improve the mechanism to determine the most suitable $K$ for different datasets in the revised version.
> > > > >
> > > > > Thanks for the time and effort in reviewing our work and we would like to express our sincere gratitude to you for providing constructive suggestions.

---

### Official Review · Reviewer_oZr8 · 2023-07-06

**Soundness:** 4 excellent
**Presentation:** 3 good
**Contribution:** 3 good
**Rating:** 7
**Confidence:** 3

**Summary:**

This paper proposes a novel framework called EAGLE for out-of-distribution generalization on dynamic graphs. EAGLE models the complex environments that influence the generation of dynamic graphs and exploits the spatio-temporal invariant patterns that can generalize under distribution shifts. EAGLE consists of four mechanisms: environment-aware dynamic graph neural network, environment instantiation, invariant pattern recognition, and causal intervention. EAGLE achieves superior performance on future link prediction tasks on both real-world and synthetic datasets compared to existing methods. The paper also provides theoretical analysis and empirical studies to support the effectiveness of EAGLE.

**Strengths:**

1. The illustration in Figure 2 is clear and aids in understanding the proposed method.
2. Theoretical analysis and proofs are provided as necessary.
3. Overall, the proposed solution for OOD Dynamic link prediction seems reasonable. The idea of generating spatio-temporal environments is novel.


**Weaknesses:**

1. The generated environment patterns lack explainability. These patterns are produced through disentangling, making it difficult to explicitly understand their specific meaning in reality.
2. The author should include further discussion on how the proposed method can contribute to link prediction in a new environment, as exemplified in Figure 1. If the model successfully distinguishes environment-invariant patterns in previous graphs, how will it capture new patterns in a different environment to improve prediction?
3. The writing style is not reader-friendly for those unfamiliar with the task and techniques. It would be beneficial to provide more intuitive explanations to enhance understanding.


**Questions:**

1. What does the one-hot multi-label y introduced in line 142 represent? Does it means what index of the environment which the observed z belongs to, or does it also include the index of the time step?
2. Could the author provide a more detailed explanation of the visualization experiment mentioned in Section 4.4?

---

> ### Author Rebuttal · Authors · 2023-08-08
>
> We sincerely thank the reviewer for the detailed comments and insightful questions. Responses are as follows.
>
> **Q1:** The generated environment patterns lack explainability.
>
> **A1:** Thank you for your question.
>
> - **[Understand latent environments]** Environments are **latent factors**, where the ground-truth labels are **inaccessible**. We provide real-world examples (lines 101-106) as illustrations. **The key insight is,** the formation of dynamic graphs typically follows a complex process under the impact of environments, causing the relationships to be multi-attribute. To effectively model diverse spatio-temporal environments, the ego-graphs of each node are disentangled and processed in **different embedding spaces**.
> - **[An example: the social networks]** The relations between a central node and its neighbors, e.g., “classmates”, “colleagues”, etc., are formed in different surrounding environments, e.g., classmates relations are formed in “school” environment, consisting of “classmates”, “teachers”, “staff”, etc., and colleagues relations are formed in “working” environment.  We propose **multi-channel environments disentangling mechanism** to discriminate different semantics in $K$ embedding spaces, e.g., the 1-st embedding space means the “classmate” relations, the 2-nd embedding space means the “colleague” relations, etc. Thus, EA-DGNN can percept and encode environment features into node representations.
>
> We will add further explanation in the revised version.
>
> ---
>
> **Q2:** Further discussion on how the proposed method can contribute to link prediction in a new environment in Figure 1. How will the model capture new patterns in a different environment?
>
> **A2:** Thank you for your question.
>
> - **[Further discussions]** We have provided further explanations in Appendix F.1. It indicates that the existing model has captured the spurious correlations between the semantics of “coffee” and “cold drink”, which caused the **false** prediction of “Iced Americano”. EA-DGNN can learn multiple environment patterns, such as  “coffee”, “cold drinks”, “iced dessert”, “summer dressing”, etc. Section 3.3 identifies the **spatio-temporal invariant pattern** to be “coffee”. Section 3.4 encourages the model to learn solely with the “coffee” pattern while minimizing the environment extrapolation risks to enhance the generalizing ability in a new environment.
> - **[During testing]** Our trained model will directly give a higher possibility score for a Hot Latte than an Iced Americano in testing. This is because the model has fully learned the invariant and sufficient information for correct prediction during training, and the risk in the unknown test environment is reduced. Besides, the model can perceive the testing environment's (winter) semantics through message passing and aggregation of its neighbor nodes, which leads to the final prediction.
>
> We will add further explanations in the revision.
>
> ---
>
> **Q3:** The writing style is not reader-friendly. Beneficial to provide more intuitive explanations.
>
> **A3:** Thank you for your suggestion. We will add more related works and further intuitive explanations to improve readability, especially to the techniques and theories. In addition, we will try to simplify the notations and add clearer clarifications to enhance understanding.
>
> ---
>
> **Q4:** What does the one-hot multi-label $\mathbf{y}$ introduced in line 142 represent? Does it mean what index of the environment which the observed $\mathbf{z}$ belongs to, or does it also include the index of the time step?
>
> **A4:** Thank you for your questions.
>
> - **[Meaning]** $\mathbf{y}$ is mixed up with time index $t$ and environment index $k$, indicating which environment index under which time index $\mathbf{z}$ belongs. It can be seen as our inferred label of environments.
> - **[Example]** If there exist $K$ environments and $T$ graph snapshots, we first initialize a zero matrix of the shape $K \times T$. Then we mark the value in position $(k,t)$ to be 1 and reshape the matrix into the 1-dimension vector $\mathbf{y}$, indicating the multi-label of $\mathbf{z}$ for the $k$-th environment at time $t$.
> - **[Role]** $\mathbf{z}$ is **concatenated** with its corresponding $\mathbf{y}$ to realize conditional variational inference for the distribution of latent environments by ECVAE. We can then instantiate environments by generating samples from the inferred distribution with any given $\mathbf{y}$. This can be regarded as **the data augmentation** under the guidance of the inferred prior distributions, which helps improve the generalization ability.
>
> We will add more explanations in the revision.
>
> ---
>
> **Q5:** Provide a more detailed explanation of the visualization experiment.
>
> **A5:** Thank you for your question.
>
> - **[Settings]** We visualized snapshots in the dataset COLLAB using NetworkX in Figure 5. As EAGLE concentrates on solving **node-level** downstream tasks, we carried out visualization analysis from the perspective of a single node $v$ (red color).
> - **[Process]** EA-DGNN first models the environment-aware node representation for node $v$ (represented by “🟪🟦🟩🟧🟨”), then Section 3.3 identifies spatio-temporal variant patterns (represented by “⛝”). To encourage the model to rely on sufficient and invariant information to make predictions and minimize the environment extrapolation risks, Section 3.4 randomly replaces the “⛝” parts with observed or generated environment samples (represented by “🔄”).
> - **[Analysis]** Thus, the model focuses more on the neighbor nodes in the **more invariant environments patterns** (purple and blue parts “🟪🟦”), increasing the edge weights (can be seen as spatio-temporal attention), while ignoring the links with the **more variant environments patterns** (represented by “🟩🟧🟨”).
>
> In conclusion, EAGLE can effectively learn invariant patterns, making generalized predictions. We will add more detailed explanations of the visualization experiment in the revision.

---

### Decision · Program_Chairs · 2023-09-21

**Decision:**

Accept (poster)

**Comment:**

This paper proposes a novel environment-aware dynamic graph learning framework for out-of-distribution generalization. In particular, the proposed framework models complex coupled environments and exploits spatial-temporal invariant patterns. Reviewers agreed that this paper studies an important problem, the proposed framework is novel, and extensive experiments are convincing. Meanwhile, reviewers raised some concerns regarding the concepts of environments, discussions of related work, and paper writing. The authors' rebuttal has successfully addressed most of the concerns. Also, the authors are strongly encouraged to incorporates the suggestions from reviewers to their final version.